# Retinoids stored locally in the lung are required to attenuate the severity of acute lung injury in male mice

Igor O. Shmarakov[1,2] ✉, Galina A. Gusarova[1], Mohammad N. Islam [1], María Marhuenda-Muñoz[1,3,4], Jahar Bhattacharya [1] & William S. Blaner [1]

Retinoids are potent transcriptional regulators that act in regulating cell proliferation, differentiation, and other cellular processes. We carried out studies in male mice to establish the importance of local cellular retinoid stores within the lung alveolus for maintaining its health in the face of an acute inflammatory challenge induced by intranasal instillation of lipopolysaccharide. We also undertook single cell RNA sequencing and bioinformatic analyses to identify roles for different alveolar cell populations involved in mediating these retinoid-dependent responses. Here we show that local retinoid stores and uncompromised metabolism and signaling within the lung are required to lessen the severity of an acute inflammatory challenge. Unexpectedly, our data also establish that alveolar cells other than lipofibroblasts, specifically microvascular endothelial and alveolar epithelial cells, are able to take up lipoprotein-transported retinoid and to accumulate cellular retinoid stores that are directly used to respond to an acute inflammatory challenge.

The literature points to linkages between retinoids (vitamin A, its natural metabolites and synthetic analogs) and lung physiology and pathophysiology[1,2]. Retinoids are potent transcriptional regulators that directly modulate the expression of >500 diverse genes, many of which are involved in the regulation of cell proliferation and differentiation[3]. The transcriptional regulatory activity of retinoids is mediated by the all-*trans*-retinoic acid (ATRA) metabolite and its three distinct cognate nuclear hormone receptors, the retinoic acid receptors (Rarα, -β, and -γ)[4–8]. ATRA is directly synthesized from retinol by two classes of dehydrogenases, retinol dehydrogenases and retinaldehyde dehydrogenases (see Supplementary Fig. 1a for a schematic overview of retinoid metabolism). Within cells and tissues, ATRA concentrations are tightly regulated. This regulation involves the actions of ATRA-inducible cytochrome P450 species (Cyp26a1 and Cyp26b1) that catalyze the oxidative degradation of ATRA, as well as the action of lecithin:retinol acyltransferase (Lrat) which catalyzes

retinyl ester formation from retinol, limiting retinol availability for ATRA synthesis. Animal model and cell culture studies have identified specific actions of ATRA/Rars in pre- and post-natal lung development and in alveoli formation in the adult lung[9–19]. Within the lung, ATRA-Rar actions are reported to be important in several lung cell types including alveolar cells, especially the type I and type II epithelial cells[10,20–26], endothelial cells[27], smooth muscle cells[11,28], and macrophages[29–31].

We now report studies that have a broad focus on retinoid metabolism and actions across a disease process, specifically experimentally induced acute lung injury (ALI), and on the roles of retinoids in different cell types affected in this disease process. There is little information available regarding either retinoid uptake into lung cells or local retinoid storage, metabolism, and actions in the context of the different cell types that are central to normal alveolar physiology and pathophysiology (Supplementary Fig. 1b). Using a well-characterized

[1]Department of Medicine, Vagelos College of Physicians and Surgeons, Columbia University, New York, NY 10032, USA. [2]Department of Animal Sciences, School of Environmental and Biological Sciences, Rutgers, The State University of New Jersey, New Brunswick, NJ 08901, USA. [3]Centro de Investigación Biomédica en Red Fisiopatología de la Obesidad y la Nutrición (CIBEROBN), Instituto de Salud Carlos III, 28029 Madrid, Spain. [4]Department of Nutrition, Food Science and Gastronomy, School of Pharmacy and Food Sciences and XIA, Institute of Nutrition and Food Safety (INSA-UB), University of Barcelona, 08921 Santa Coloma de Gramenet, Spain. ✉e-mail: ishmarakov@sebs.rutgers.edu

mouse model for ALI, we have addressed this lack of information. We hypothesized that vascular endothelial cells within vitamin A-responsive tissues like the lung play a very significant and previously unrecognized role in the uptake and metabolism of vitamin A. One overall objective of our investigations was to gain better understanding of how retinoid metabolism within the adult lung, upstream of the actions of ATRA and the Rars, influences the development of acute lung disease. The surprising conclusion from our studies is that local retinoid stores are present in a number of different cell types within the lungs and that the metabolism and intercellular transport of locally stored retinoids, and not retinoids directly derived from the circulation, are required for limiting ALI severity.

## Results

### Lungs contain local retinoid stores which are preferentially utilized during ALI

To understand how retinoids that are stored as retinyl esters (REs) within the lung are utilized in response to an inflammatory stimulus during ALI, we first analyzed concentrations of the two predominant retinoid species present in the lung, retinyl esters (RE) and all-*trans*-retinol (ROH), at baseline and 7 days after lipopolysaccharide (LPS) instillation using a dose of 25 mg/kg. In agreement with earlier reports[32,33], HPLC analysis of lung homogenates confirmed the presence of both of these retinoid species at relatively high concentrations in the lungs at baseline (Fig. 1a, b, Supplementary Fig. 2). Pulmonary RE and ROH concentrations for 3-month-old chow-fed male mice were $555.6 \pm 93.6$ nmol/g and $9.6 \pm 3.3$ nmol/g respectively (Fig. 1a, b). When compared to retinoid concentrations in the liver, the central organ for retinoid storage in the body, lung values were about 40% of those of hepatic RE concentrations ($1485.1 \pm 197.1$ nmol/g) but only 10% of hepatic ROH levels ($91.7 \pm 21.8$ nmol/g) (Fig. 1a, b). Interestingly, the lung retinyl ester acyl composition was different from that of the liver. Specifically, retinyl palmitate and retinyl stearate were the predominant retinyl ester species detected in almost equimolar concentrations in pulmonary homogenates (Supplementary Fig. 2). This is unlike the liver where retinyl palmitate accounts for ~75% of all RE.

Triggering ALI by instilling LPS (25 mg/kg) intranasally into mice resulted in a marked decline, within 7 days, in lung RE concentrations by >80% and lung ROH concentrations by >40% (Fig. 1a, b). The decline in pulmonary retinoids was not associated with significant decline in hepatic RE (Fig. 1a). Thus, local pulmonary RE stores were drawn upon during an acute inflammatory challenge rather than hepatic retinoid (RE) stores. Plasma ROH, a retinoid form that is available to the lungs from the blood, was almost 60% lower than in control mice instilled with PBS (Fig. 1c). For ROH to be present in the circulation, it must be secreted from the liver and must be bound to the hepatocyte-derived transport protein retinol-binding protein 4 (Rbp4)[34]. Rbp4 is a known negative acute phase response protein[35,36] and RBP4 plasma concentrations have been proposed as a predictive biomarker for mortality in patients with acute exacerbations of COPD[37]. Consequently, we checked whether Rbp4 protein levels change in plasma in response to the LPS challenge. Indeed, 7 days after 25 mg/kg LPS instillation, the acute phase response evidenced by elevated plasma C-reactive protein concentrations in wild-type mice (Supplementary Fig. 3a) was associated with an ~40% decline in plasma Rbp4 levels (Fig. 1d) and significantly downregulated hepatic *Rbp4* mRNA and protein expression, each by ~40% (Supplementary Fig. 3b, c). These observations are consistent with what is known for infections and associated inflammatory responses in humans which result in an ~40–50% lowering of plasma retinol and RBP4 levels[38–41]. Rodent studies, including ones involving inflammation induced by LPS[36,42–44] have established that the observed declines arise primarily from diminished hepatic Rbp4 synthesis and secretion[36,42–46]. Since retinol can only be mobilized from the liver bound to Rbp4, this accounts for the lower plasma retinol levels. This literature further establishes that neither renal

filtration nor tissue uptake and catabolism contribute significantly to the decline in circulating retinol-Rbp4. Based on this literature, we suggest that the decline in plasma ROH levels we observed likely arose due to lessened retinol-Rbp4 secretion from the liver as opposed to increased utilization of circulating retinol by inflamed tissues. However, this will need to be shown experimentally before this can be considered to be definitive.

### Local pulmonary retinoid stores, but not circulating ROH are critical for survival during ALI

To investigate the relative importance of local versus peripheral retinoid sources in facilitating lung responses to acute inflammatory stress, we employed two previously characterized mouse models of retinoid-insufficiency[34,47]. One lacks the ability to store retinoids, either locally in the lung or within the liver, due to the deletion of the gene encoding lecithin:retinol acyltransferase (*Lrat*-deficient or *Lrat*$^{-/-}$ mice)[47]. *Lrat* encodes a protein that is responsible for the preponderance of RE synthesis in most tissues including the liver and lungs[47,48]. Lack of Lrat, due to genetic ablation of the *Lrat* gene, results in an almost complete inability to synthesize REs in these tissues, resulting in nearly undetectable lung RE concentrations for *Lrat*$^{-/-}$ mice (Supplementary Fig. 4a). Because the ROH that is being consumed from the diet cannot be stored in the tissues of *Lrat*$^{-/-}$ mice, including the lungs, it ultimately undergoes enzymatic oxidation (as outlined in Supplementary Fig. 1a) that has been associated with higher tissue ATRA concentrations. Normally, the elevated ATRA concentrations are efficiently eliminated via a feedback mechanism involving upregulation of *Cyp26a1* and *Cyp26b1* (Supplementary Fig. 4b), ATRA-responsive hydroxylases that specifically oxidize ATRA[49]. As a result of this metabolic adaptation, *Lrat*$^{-/-}$ mice have nearly undetectable levels of REs in most tissues, including lungs (Supplementary Fig. 4a). However, pulmonary and plasma ROH concentrations in *Lrat*$^{-/-}$ mice are about 10 and 50% respectively of those present in matched wild-type animals (*Lrat*$^{+/+}$ mice) (Supplementary Fig. 4c, d). Under normal physiological conditions when maintained on a retinoid-sufficient diet, *Lrat*$^{-/-}$ mice are capable of synthesizing sufficient amounts of ATRA from retinol and thus are physiologically normal with no observable adverse phenotypic consequences[47,50]. The other model is one that possesses no circulating Rbp4 (*Rbp4*-deficient or *Rbp4*$^{-/-}$ mice[34]) and therefore has nearly undetectable levels of plasma ROH (Supplementary Fig. 4e). Since Rbp4 is required for mobilizing hepatic retinoid stores, the absence of *Rbp4* results in an inability of *Rbp4*$^{-/-}$ mice to mobilize ROH from the liver to meet the retinoid demands of peripheral tissues like the lung. The extrahepatic tissues of *Rbp4*$^{-/-}$ mice acquire and accumulate retinoids delivered postprandially by intestinal lipoproteins[34,51]. Our data show that pulmonary retinoid concentrations in *Rbp4*$^{-/-}$ mice are not different from those of littermate controls (*Rbp4*$^{+/+}$ mice) (Supplementary Fig. 4f, g), indicating that retinoid delivery to the lungs by circulating ROH-Rbp4 does not normally play a large role in the ability of the lungs to acquire and accumulate retinoids. Through the study of these two animal models we were able to recapitulate the conditions of either local pulmonary retinyl ester absence but with available circulating ROH (*Lrat*$^{-/-}$ mice), or very low circulating retinol levels with normal lung RE stores (*Rbp4*$^{-/-}$ mice).

Both *Lrat*$^{-/-}$ and *Rbp4*$^{-/-}$ mice, as well as their respective littermate controls (*Lrat*$^{+/+}$ and *Rbp4*$^{+/+}$ mice), were dosed with LPS (25 mg/kg) via intranasal instillation, and survival was monitored for 7 days. Starting from day 3, the survival rate of *Lrat*$^{-/-}$ mice began to decline, with only 15% surviving through day 7, as opposed to almost 80% survival of the *Lrat*$^{+/+}$ mice (Fig. 1e). There was a statistically significant difference with $p = 0.0004$ between the survival curves for LPS-treated *Lrat*$^{+/+}$ and *Lrat*$^{-/-}$ mice as calculated by the Kaplan–Meier log rank test. The survival rate of LPS-instilled *Rbp4*$^{-/-}$ mice was not significantly different from wild type (*Rbp4*$^{+/+}$) controls, at a level of

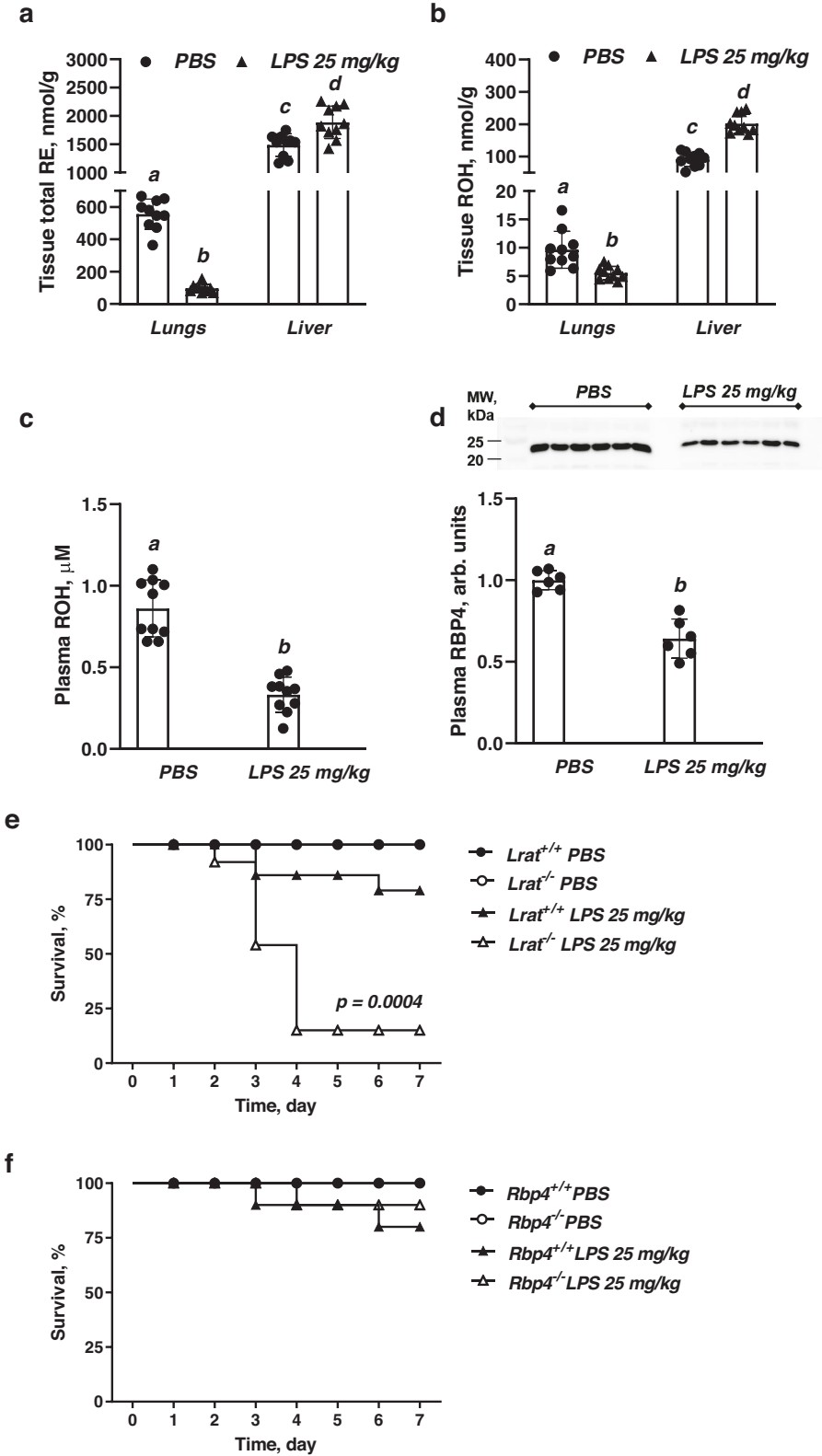

80% (Fig. 1f). Thus, local pulmonary retinoid stores, but not circulating ROH, are critical for assuring a better ALI outcome.

## Retinoid storage in the lung is mapped to distinct pulmonary cell types

To gain deeper insight into cellular aspects of retinoid storage in the lungs, we isolated and collected retinoid-containing cells (referred to as UV-positive cells) from lung by FACS making use of their autofluorescence and emission at 455 nm upon excitation at 350 nm (Fig. 2a). The autofluorescence of cellular retinoid stores was first described in the early 1980s[52–54]. This autofluorescence is very distinctive, since it involves excitation at 350 nm but the characteristic blue-green emission occurs at 450–460 nm. This unusual characteristic (specifically, the considerable distance between the excitation and

**Fig. 1 | Tissue retinoid concentrations and survival of mice during LPS-induced ALI. a** Total retinyl ester concentrations (nmol/g) in lungs and liver determined by HPLC in mice 7 days after intranasal instillation of LPS (25 mg/kg of body weight in PBS) or PBS alone. Values marked with different letters (a, b, c, d) are statistically different (a is different from b, $p = 1.80e\text{-}06$; c is different from d, $p = 1.67e\text{-}05$). Statistical differences were first analyzed by a one-way ANOVA followed by multiple comparisons employing Tukey's HSD *post hoc* test. All values are given as the mean $\pm 1$ S.D., $n = 10$ for each group. **b** Retinol concentrations (nmol/g) in lungs and liver determined by HPLC in mice 7 days after intranasal instillation of LPS (25 mg/kg of body weight in PBS) or PBS alone. Values marked with different letters (a, b, c, d) are statistically different (a is different from b, $p = 0.0014$; c is different from d, $p = 1.92e\text{-}08$). Statistical differences were first analyzed by a one-way ANOVA followed by multiple comparisons employing Tukey's HSD post hoc test. All values are given as the mean $\pm 1$ S.D., $n = 10$ for each group. **c** Plasma retinol concentrations ($\mu$M) determined by HPLC in mice 7 days after intranasal instillation of LPS (25 mg/kg of body weight in PBS) or PBS alone. Values marked with different letters (a, b) are statistically different (a is different from b, $p = 1.91e\text{-}07$). Statistical differences were analyzed by a one-way ANOVA. All values are given as the mean $\pm 1$ S.D., $n = 10$ for each group. **d** Quantitative representation of retinol-binding protein 4 (Rbp4) concentrations determined by immunoblot (upper insert) in plasma of mice 7 days after intranasal instillation of LPS (25 mg/kg of body weight) or vehicle. Each lane represents an extract prepared from an individual animal. Values marked with different letters (a, b) are statistically different (a is different from b, $p = 6.23e\text{-}05$). Statistical differences were analyzed by a one-way ANOVA. All values are given as the mean $\pm 1$ S.D., $n = 6$ for each group. **e, f** Kaplan–Meier survival curves for *Lrat*[−/−] and *Rbp4*[−/−] mice as well as for their respective littermate controls (*Lrat*[+/+] (**e**) and *Rbp4*[+/+] (**f**)) over the 7-day period after intranasal instillation of a LPS (25 mg/kg of body weight in PBS) or PBS alone. A statistically significant difference ($p = 0.0004$) was detected using the log rank test comparing the survival curves for LPS-instilled *Lrat*[+/+] and *Lrat*[−/−] mice, $n = 14$ for each group.

emission maxima) for this autofluorescence provides confidence in the use of FACS to isolate retinoid-storing cells. Retinoid autofluorescence has been used in published studies to establish the identities of retinoid-storing cells and/or to quantitate the abundance of these cells[32,55–57]. We[58] and others[59–62] have used this characteristic to isolate by FACS and study retinoid-storing cells from living tissues. It has long been known from the literature that a subpopulation of pulmonary mesenchymal stromal cells with fibroblastic characteristics, referred to as pulmonary lipid interstitial cells or pulmonary lipofibroblasts, are capable of accumulating lipids, including REs[63,64]. We used this knowledge to isolate and further characterize retinoid-accumulating cells from the lung. Lungs from *Lrat*[+/+] and *Lrat*[−/−] mice were enzymatically digested and individual cells were subjected to FACS using a gating strategy to collect cells with high autofluorescence at $\lambda = 455$ nm and negative staining with propidium iodide (i.e., viable cells) (Fig. 2a). Using this procedure, we were able to collect about $5 \cdot 10^5$ UV-positive cells from one wild type (*Lrat*[+/+]) lung. Notably, for cell isolates from *Lrat*[−/−] lungs, which contain no REs, we were not able to detect cells with high autofluorescence at $\lambda = 455$ nm (UV-positive cells) using the identical FACS gating strategy (Fig. 2b). This confirms the specificity of this protocol for isolating retinoid-containing cells.

In culture, a majority of the freshly isolated UV-positive cells had an elongated quiescent fibroblast morphology with distinct intracellular lipid droplets (Fig. 2c). HPLC analysis of the isolated cells allowed for the identification of ROH and all RE species, including retinyl linoleate, retinyl oleate, retinyl palmitate, and retinyl stearate (Fig. 2d, Supplementary Fig. 5a). After in vitro culture for 3 days, the cells acquired a spindle-shaped morphology with long cytoplasmic extensions with fewer intracellular inclusions (Fig. 2c). Cellular RE concentrations declined during the 3-day in vitro culture period and this was accompanied by downregulated expression of *Lrat* (Fig. 2e). In addition, the cultured cells acquired the ability to express strongly the fibrosis marker smooth muscle actin (*Acta2*) (Fig. 2f).

To identify the specific cell types and associated transcriptional signatures of pulmonary retinoid-containing cells, the UV-positive cells we collected were subjected to single-cell RNA sequencing (scRNA-seq). Post-processing analysis of scRNA-seq data from integrated datasets identified a total of 10 clusters (clusters 0-9) of cells using cluster resolution 0.2 (in *FindClusters* function) (Fig. 3a, Supplementary Fig. 6a, b). The cluster resolution value was determined using the *Clustree* function (Supplementary Fig. 6c). Biologically relevant cell clusters were defined as clusters that have at least 10 unique differentially expressed genes. Visualization of the clusters on a 2D map was performed using uniform manifold approximation and projection (UMAP) (Fig. 3a, Supplementary Fig. 6a). Unexpectedly, pulmonary fibroblasts were not the only lung cell type capable of accumulating retinoids, as evidenced by FACS sorting based on retinoid autofluorescence. Using the expression of representative highly discriminative tissue-specific markers, including collagen type I alpha 1

chain (*Col1a1*), epithelial cell adhesion molecule (*Epcam*), platelet and endothelial cell adhesion molecule 1 (*Pecam1*, also known as *Cd31*), and protein tyrosine phosphatase receptor type C (*Ptprc*, also known as *Cd45*), clusters corresponding to four major cell groups were identified as harboring retinoid-stores (Fig. 3a, b). Among the UV-positive cell clusters, the stromal cell group (83% of total collected cells) was the most abundant. This group consisted of 3 distinct but related cell clusters (clusters 0, 1, and 2) that were enriched with general fibroblast marker transcripts, including *Col1a1*, *Pdgfra*, and *Ptgis* (Fig. 3c). A distinct cluster of endothelial cells (about 7% of total cells, cluster 3) was identified based on its enrichment with *Pecam1* transcripts (Fig. 3b) as well as transcripts for other endothelial marker genes, including cadherin 5 (*Cdh5*) and plasmalemma vesicle-associated protein (*Plvap*) (Fig. 3c). An epithelial cell group (cluster 4) constituted about 5% of total cells. The cells from this cluster were expressing mostly marker genes for alveolar type 2 cells (AEC2), including surfactant protein genes *Sfpta1* and *Sfptb* (Fig. 3c) as well as *Sftpc* and *Abca3*[65,66] (Supplementary Fig. 6d). The cells from this group were also expressing certain markers attributed to transcriptional signatures of alveolar type 1 cell (AEC1), including *Aqp5* and *Ager*. However, these cells did not express other markers of AEC1 late differentiation[65–67] like *Pdpn, Cav1, Hopx, Scnn1g, and Scnn1b* (Supplementary Fig. 6d). Myeloid cells were represented by 5 clusters (about 5% of total cells, clusters 5-9) differentially expressing immune cell marker genes, including interleukin-1β (*Il-1β*) and lymphocyte-specific protein 1 (*Lsp1*) (Fig. 3c).

Consistent with our scRNA-seq data, the presence of endothelial, epithelial, and myeloid cells among retinoid-containing UV-positive pulmonary cells was further corroborated by FACS using antigen-specific fluorochrome-labeled antibodies (Supplementary Fig. 7). Cd45[+] (myeloid), Cd326[+] (epithelial), and Cd31[+] (endothelial) cells were identified among sorted UV-positive cells (Supplementary Fig. 7) and collected for further analysis. HPLC analysis of retinoids present in the collected cell preparations allowed us to identify detectable amounts of retinyl esters in each cell type. Clear peaks corresponding to retinyl palmitate and retinyl stearate were identified in Cd31[+] (endothelial) cells (Supplementary Fig. 5b), and retinyl palmitate in Cd326[+] (epithelial) cells (Supplementary Fig. 5c).

Taken together these data clearly show that several lung cell populations are capable of accumulating retinoids in a cell-specific manner and this feature can be reproducibly utilized for their specific selection and isolation.

## Pulmonary cells express genes needed for retinoid metabolism and signaling in a cell-specific manner

Since cells were FACS sorted and collected based on their endogenous retinoid-fluorescence, we analyzed for expression of genes controlling retinoid uptake, accumulation, and metabolism among the different cell clusters identified in our dataset. Strong mRNA expression of *Lrat* and retinol-binding protein 1 (*Rbp1*), a

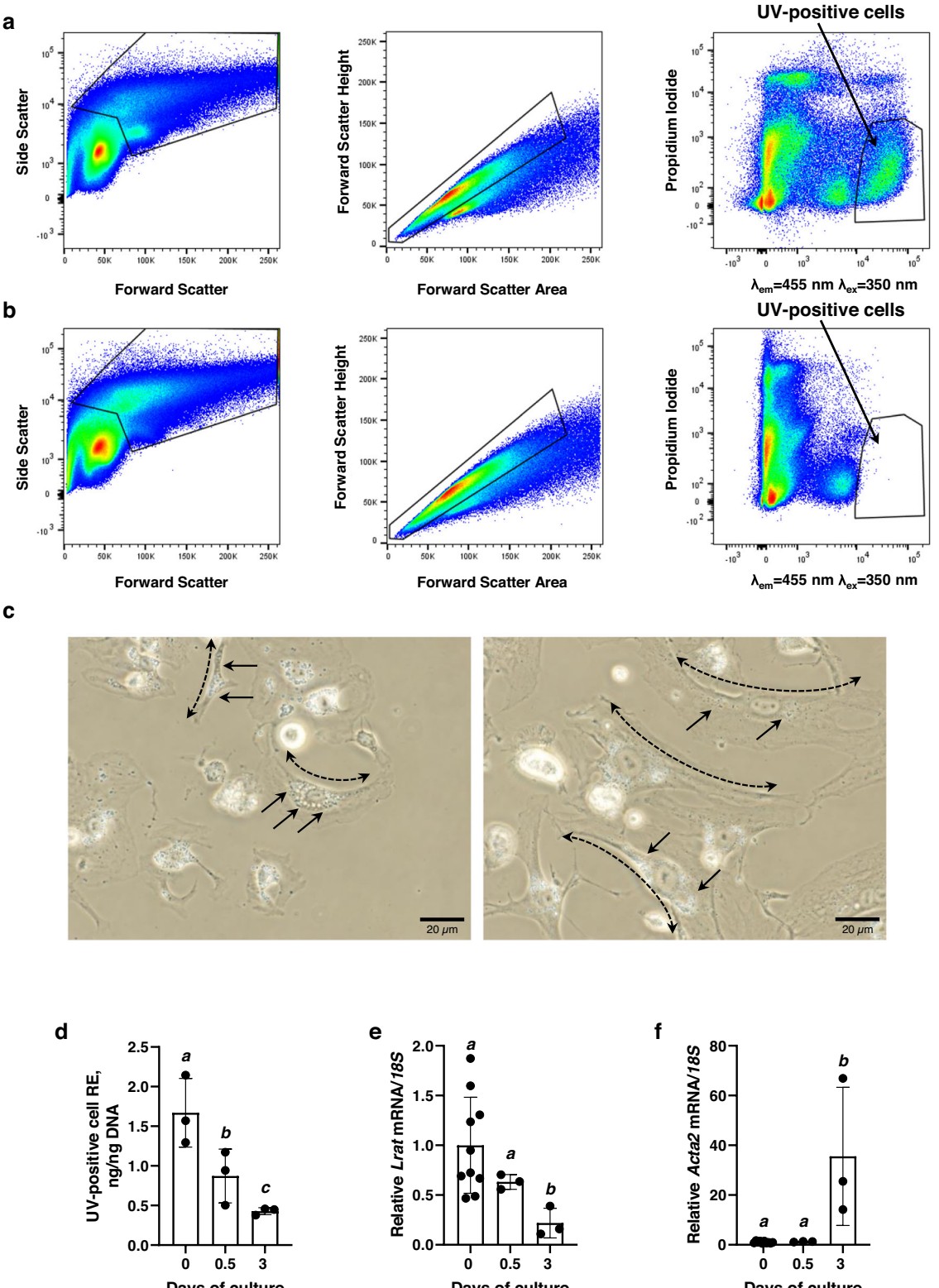

gene encoding a protein needed for intracellular transport of the hydrophobic ROH, facilitating both ROH esterification by Lrat or its oxidation to ATRA, was detected predominantly in the stromal (clusters 0–2) and endothelial (cluster 3) cell clusters (Fig. 3d). These scRNA-seq data were further confirmed by qRT-PCR and immunoblot analyses using whole lung homogenates and primary lipofibroblasts and endothelial cells (Fig. 3e, f). In addition, transcriptional signals for both *Lrat* and *Rbp1* were detected in

cells from the epithelial and myeloid cell clusters, albeit at lower levels (Fig. 3d).

Further analysis of scRNA-seq datasets generated in our experiments allowed us to identify cell-specific expression of genes involved in ATRA synthesis and signaling (Fig. 3d, g). Specifically, the expression of retinol dehydrogenases (*Rdh10*, *Rdh11*, *Rdh12*), genes encoding enzymes that catalyze retinol oxidation to retinaldehyde, a first enzymatic step towards ATRA generation[68–70], was detectable in all cell

**Fig. 2 | FACS isolation and culture of lung retinoid-containing cells.** A gating strategy for sorting live, single, UV-positive cells (defined by the enclosed area) using emission at λ = 455 nm upon excitation at λ = 350 nm from lung cell suspensions isolated from *Lrat*[+/+] (**a**) and *Lrat*[−/−] (**b**) mice. **c** Isolated retinoid-containing (UV-positive) cells cultured 12 h (left microphotograph, 40X magnification) or 72 h (right microphotograph, 40X magnification); Dashed arrows indicate elongated cell morphology and solid arrows highlight intracellular lipid droplets. **d** Total retinyl ester concentrations (ng/ng DNA) present in UV-positive cells determined by HPLC after 0, 0.5 and 3 days of culture. Values marked with different letters (a–c) are statistically different (a is different from b, $p = 0.0222$; a is different from c, $p = 0.0031$). Statistical differences were first analyzed by a one-way ANOVA followed by multiple comparisons employing Tukey's HSD post hoc test. All values are

given as the mean ± 1 S.D., $n = 3$ for each group. **e** *Lrat* mRNA expression determined by qRT-PCR after 0, 0.5 and 3 days of culture. Values marked with different letters (a, b) are statistically different (a is different from b, $p = 0.012$). Statistical differences were first analyzed by a one-way ANOVA followed by multiple comparisons employing Tukey's HSD post hoc test. All values are given as the mean ± 1 S.D., $n = 3$ for each group, $n = 10$ for *Lrat* mRNA expression on day 0. **f** *Acta2* mRNA expression determined by qRT-PCR after 0, 0.5 and 3 days of culture. Values marked with different letters (a, b) are statistically different (a is different from b, $p = 0.002$). Statistical differences were first analyzed by a one-way ANOVA followed by multiple comparisons employing Tukey's HSD post hoc test. All values are given as the mean ± 1 S.D., $n = 3$ for each group, n = 10 for *Acta2* mRNA expression on day 0.

types identified in our dataset (Fig. 3d). On the other hand, expression of *Dhrs3*, a gene encoding reductase that catalyzes the reverse reaction of retinaldehyde conversion to retinol[71], was the highest in two stromal (clusters 0 and 1), endothelial (cluster 3), epithelial (cluster 4), and one myeloid (cluster 5) cell clusters (Fig. 3d). Among different aldehyde dehydrogenases (*Aldh1a1*, *Aldh1a2*, *Aldh1a3*) which catalyze the irreversible oxidation of retinaldehyde to ATRA, *Aldh1a1* was the most highly expressed isoform. *Aldh1a1* was specifically expressed in stromal (clusters 0–2), endothelial (cluster 3) and one myeloid (cluster 5) cell clusters. The two cellular retinoic acid-binding proteins (*Crabp1* and *Crabp2*) that channel ATRA towards either oxidation or nuclear translocation were expressed at low, but detectable levels mostly in stromal (clusters 0, 1, and 2), endothelial (cluster 3) and epithelial (cluster 4) cell clusters. Detectable levels of *Cyp26b1* transcripts encoding one cytochrome P450 isoform that catalyzes substrate-specific catabolic oxidation of ATRA were detected in the majority of cell clusters, except for myeloid cell clusters 7, 8, and 9 (Fig. 3d). No transcripts for *Cyp26a1* were detected in our scRNA-seq datasets.

mRNA expression of canonical retinoic acid receptors (*Rarα*, *Rarβ*, *Rarγ*) that mediate ATRA signaling as well as their heterodimerization partners (*Rxrα*, *Rxrβ*, *Rxrγ*) was detected in all cell clusters with distinctive cell-specific patterns of expression (Fig. 3g). Specifically, *Rarα* was mostly expressed in endothelial (cluster 3) and 2 myeloid (clusters 5 and 6) cell clusters; high *Rarγ* expression was only attributed to one of the stromal (cluster 1) and endothelial (cluster 3) cell clusters, while *Rarβ*, the least expressed *Rar* isoform, was detectable in most of the retinoid-containing cells, except for three myeloid cell clusters (Fig. 3g).

Surprisingly, one of the pulmonary stromal clusters (cluster 1) expressed *Rbp4* (Fig. 3h), which is needed for ROH transport through both plasma and the extracellular space[34]. Rbp4 is secreted predominantly by hepatocytes[34,72,73].

To assess whether cells composing the different clusters can acquire retinoids in the form of either ROH or preformed REs from extracellular sources, a number of genes encoding proteins involved in retinoid uptake were screened (Fig. 3h). Consistent with the notion that lung cells do not rely substantially on ROH delivered to the tissue by plasma Rbp4, none of cells that compose the 10 cell clusters expressed the gene encoding stimulated by retinoic acid 6 (*Stra6*)[74], a cell surface receptor for Rbp4. However, a number of different cell clusters expressed genes encoding proteins involved in retinoid uptake from lipoproteins. The gene encoding lipoprotein lipase (*Lpl*), which is needed for retinoid uptake from lipoproteins[75], was expressed exclusively by one stromal cluster (cluster 1) as well as endothelial, epithelial, and one immune cell (cluster 5) clusters. Glycosylphosphatidylinositol-anchored high-density lipoprotein binding protein 1 (*Gpihbp1*) which mediates the transport and anchoring of Lpl in capillaries[76] was highly expressed in endothelial cells (cluster 3). The fatty acid uptake transporter *Cd36* was another highly expressed lipid uptake gene detected in these retinoid-containing endothelial cells, epithelial, and myeloid cell clusters 5 and 6 (Fig. 3h). All of the cell clusters, except for endothelial cluster 3

and myeloid cluster 6, had low levels of transcripts for scavenger receptor class B type 1 (*Scarb1*), a cell surface receptor for different lipid ligands. The very low-density lipoprotein receptor (*Vldlr*) gene was highly expressed in most of the fibroblast clusters, endothelial, epithelial, and myeloid cluster 5, while transcripts for low-density lipoprotein receptor (*Ldlr*) were detected only in epithelial and one immune cell cluster (cluster 6). Low-density lipoprotein receptor-related protein 1 (*Lrp1*), an endocytic cell surface receptor proposed to mediate retinoid uptake in the placenta and fetal lungs[77,78], was highly expressed in all fibroblast clusters but less strongly by endothelial and selected immune cell clusters (clusters 5 and 6). Expression of activin-like kinase 1 (*Acvrl1*), a gene encoding protein that mediates low density lipoprotein (LDL) uptake into endothelial cells[79], was high in endothelial (cluster 3), but also in stromal (clusters 1–3) and one myeloid (cluster 5) cell clusters. High levels of mRNA expression in endothelial (cluster 3) and one stromal cluster (cluster 2) were also detected for *Cav1*, a gene encoding caveolin-1 that regulates transcytosis of LDL across endothelial cells and plasma LDL levels[80]. Only one myeloid cell cluster (cluster 9) expressed mRNA for diacylglycerol O-acyltransferase 1 (*Dgat1*), an enzyme that possesses retinol acyltransferase activity (Fig. 3h) and which can account for some RE synthesis when very high levels of retinoids are present in the gut[81]. Expression of diacylglycerol O-acyltransferase 2 (*Dgat2*), an acyltransferase capable of catalyzing RE synthesis, was low across all the cell clusters in our dataset (Fig. 3h).

Taken together, our scRNA-seq data established that distinct lung cell types of different developmental origins, including stromal, endothelial, epithelial, and myeloid cells, express genes needed for retinoid uptake, storage, metabolism, and signaling and that consequently can accumulate detectable levels of intracellular retinoids. The predominance of pulmonary mesenchymal stromal cells among isolated retinoid-containing cells and their ability to metabolize retinoids is consistent with the literature[63,64]. However, the unexpected identification of endothelial, epithelial, and myeloid cells among the pulmonary retinoid-containing cells along with the transcriptional signatures underlying retinoid uptake, accumulation, and metabolism suggests that subpopulations of these cells have actions in pulmonary retinoid metabolism.

To understand better the heterogeneity of pulmonary retinoid-containing fibroblasts and to annotate specific cell types to the stromal clusters, we removed from further analysis all non-stromal clusters and re-clustered the stromal clusters (Fig. 4a). Cluster resolution 0.2 was applied in the *FindClusters* function, which was determined using the *Clustree* function (Supplementary Fig. 6e). Biologically relevant cell clusters were defined as clusters that have at least 10 unique differentially expressed genes. Visualization of the clusters on a 2D map was performed using uniform manifold approximation and projection (UMAP) (Fig. 4a). We then carried out a differential gene expression analysis and identified expression markers for each fibroblast cluster and compared our scRNA-seq dataset to gene patterns previously reported for different pulmonary fibroblast populations (Supplementary Fig. 8)[82–86]. We were able to categorize the majority of stromal

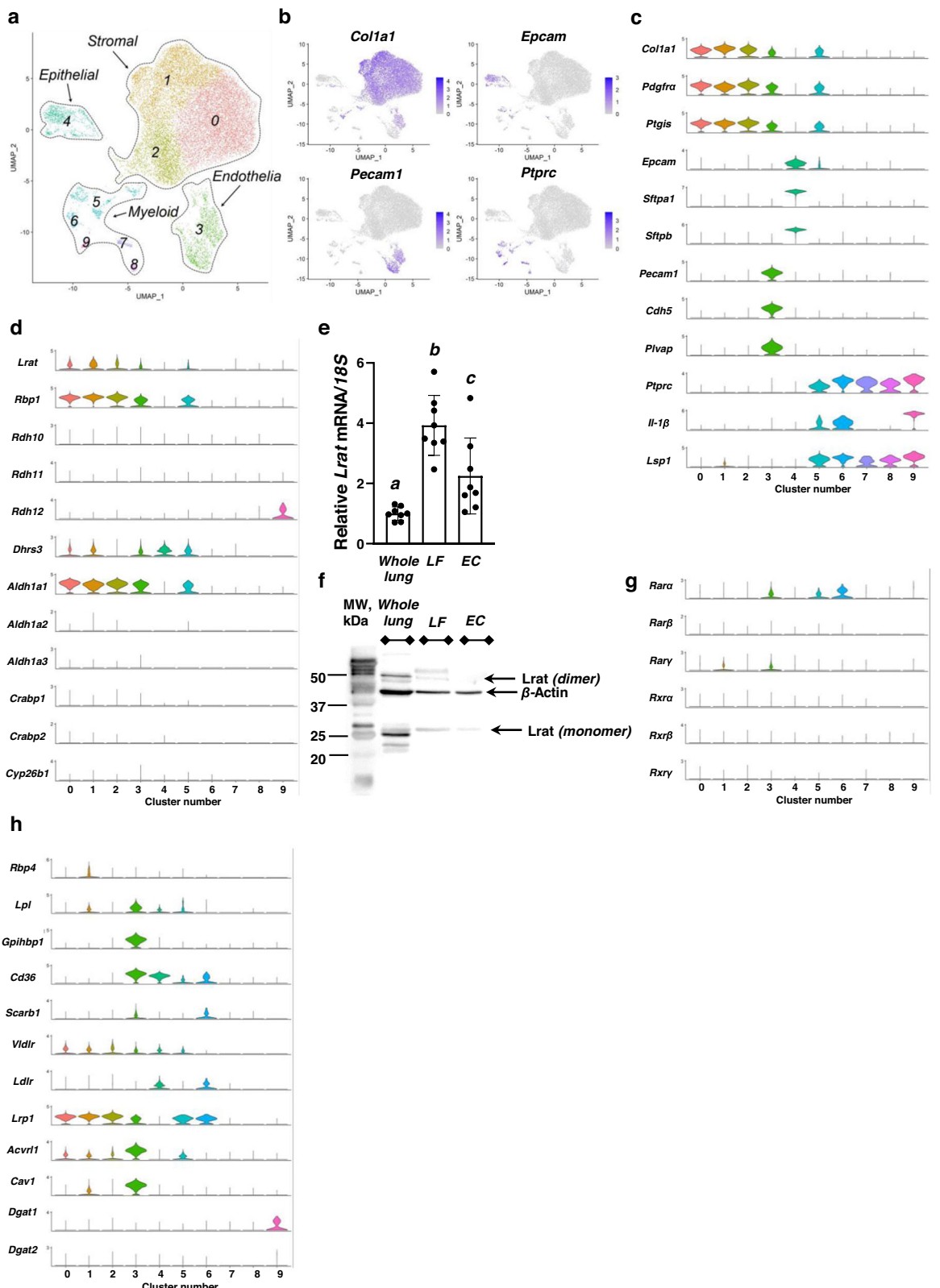

clusters identified in our study into several previously reported cell groups (Fig. 4b). The comparative analysis of the transcriptional signatures allowed us to conclude that the majority of stromal clusters identified in our dataset were enriched with markers indicative of different types of matrix fibroblasts (Fig. 4b) but depleted in markers for myofibroblasts, pericytes, fibromyocytes, mesothelial cells, and airway smooth muscle cells (Supplementary Fig. 8)[82–86]. This

transcriptional pattern allowed the clusters to be further divided into two groups: one composed of clusters 0 and 2 and the other composed of cluster 1 (Fig. 4b, 4c).

Clusters 0 and 2 shared high expression of marker genes that have previously been proposed to characterize alveolar fibroblasts[83,84] as well as *Col13a1*+ matrix fibroblasts[82,86], including *Slc7a10, Tagln, Fgfr4, Limch1, Col13a1, Pdgfrα, Npnt, Bsg, Itga8, Vegfd, Tbx2, Ptgis, Mfap4,*

**Fig. 3 | Postprocessing analysis of single-cell RNA-seq datasets of mouse lung retinoid-containing cells. a** UMAP visualization of cell type clustering inferred from mouse lung retinoid-containing cell scRNA-seq data; Ten retinoid-containing cell clusters were identified. **b** UMAP visualization of relative expression of representative canonical cell type specific markers to validate stromal (*Col1a1*), epithelial (*Epcam*), endothelial (*Pecam1*), and myeloid (*Ptprc*) cell clusters. The intensity of expression is indicated by blue coloration. **c** Violin plot representation showing known stromal (*Col1a1*, *Pdgfra*, and *Ptgis*), epithelial (*Epcam*, *Sfpta1*, and *Sfptb*), endothelial (*Pecam1*, *Cdh5*, and *Plvap*), and myeloid (*Ptprc*, *Il-1β*, and *Lsp1*) marker gene expression across all clusters. The y-axis indicates normalized Log2 expression value, the x-axis indicates cell cluster number. **d** Violin plot representation showing expression of genes involved in retinoid metabolism across all clusters. The y-axis indicates normalized Log2 expression value, the x-axis indicates cell cluster number. **e** Relative *Lrat* mRNA expression in whole tissue homogenate (whole lung), primary lipofibroblasts (LF), and primary endothelial cells (EC) isolated from the lungs of wild-type (*Lrat*$^{+/+}$) mice. Gene expression values (normalized to 18 S rRNA levels) were determined by qRT-PCR. Values marked with different letters (a, b, c) are statistically different (a is different from b, $p = 3.26e-06$; a is different from c, $p = 0.0144$; b is different from c, $p = 0.0017$). Statistical differences were first analyzed by a one-way ANOVA followed by multiple comparisons employing Tukey's HSD post hoc test. All values are given as the mean ± 1 S.D., $n = 8$ for each group. **f** Representative immunoblot of relative Lrat protein expression levels (normalized to β-actin levels) in whole tissue homogenate (whole lung), primary lipofibroblasts (LF), and primary endothelial cells (EC) isolated from the lungs of wild-type (*Lrat*$^{+/+}$) mice; similar results were reproduced in three independent experiments. **g** Violin plot representation showing expression of genes involved in retinoid signaling across all clusters. The y-axis indicates normalized Log2 expression value, the x-axis indicates cell cluster number. **h** Violin plot representation showing expression of genes involved in lipid metabolism across all clusters. The y-axis indicates normalized Log2 expression value, the *x*-axis indicates cell cluster number.

*Spon1*, and *Cdh11* (Fig. 4b). Clusters within this group, in addition to expressing known markers of alveolar fibroblasts, also highly expressed the genes that define lung lipofibroblasts (*Tcf21, Plin2, Pdgfrα, GOs2, Gyg, Wnt2, Pdzd2*)[83,86], and genes involved in retinoid and lipid metabolism (*Lrat, ApoE, Rbp1, Lpl, Ces1d, Lrp1*)[47,75,77,78], allowing us to annotate these two clusters as conventional pulmonary lipofibroblasts within the fibroblast group (Fig. 4b).

In addition to the identification of known and previously reported pulmonary lipofibroblast marker genes, analysis of differentially expressed genes in the stromal clusters allowed us to identify additional transcriptional signatures that were unique for each cluster (Fig. 4c). The majority of top highly representative genes expressed in cluster 0 were genes encoding proteins involved primarily in cell proliferation, migration, motility, and adhesion. These included genes encoding transcription factors, transcriptional activators and corepressors, and enhancer proteins that regulate proliferation in response to growth factor stimulation (*Jun, Junb, Jund, Fos, Fosb, Ier2, Egr1, Atf3, Cebpd, Btg2, Klf6*) as well as proteins regulating transduction of growth factor signaling (*Ccn1, Ccn2, Rhob, Ppp1r15a, Socs3, Dusp1, Sat1*). Taken together, these transcriptional signatures suggest that cluster 0 lipofibroblasts are cells with a high proliferative potential and mobility or "proliferative" lipofibroblasts. Cluster 2 fibroblasts, which together with cluster 0 cells share the expression signature of pulmonary lipofibroblasts, did not express the proliferative genes seen for cluster 0. Rather, the fibroblasts from cluster 2, in addition to being enriched in lipofibroblast-specific transcripts, highly expressed genes encoding proteins needed for cell adhesion (*Pdzd2*), growth (*Igfbp3, Cdkn2c*), differentiation (*Tagln, Hopx*), amino acid uptake (*Slc7a10*), and lipid hydrolysis (*Ces1d*). These data clearly establish that even within the pulmonary lipofibroblast population, there is cellular diversity in the commitment of cells to retain the proliferative vs the differentiated state.

On the other hand, cluster 1 cells shared high expression levels of marker genes attributed to adventitial fibroblasts[83,84], as well as *Col14a1*$^+$ fibroblasts[82,84]. These markers included *Col1a1, Col1a2, Igfbp7, Bgn, Gpc3, Eln, C3, Mfap5, Fstl1, Dcn, Dkk3, Cygb, Mmp3, Heyl*, and *Col14a1* (Fig. 4b, Supplementary Fig. 8). Cluster 1 was the one expressing most of the unique fibroblast genes, including *Tm4sf1, Adh7, Pi16*, and *Serpinf1* (Fig. 4b). Interestingly, the gene encoding *Rbp4* was amongst the most highly discriminatory marker genes characterizing this cluster (Fig. 4b, c). Cluster 1 also comprised cells expressing genes that some investigators associate with myofibroblasts and/or pericytes[83–85], including *Crip1, Htra1, Pdgfrβ, Acta2, Tgfβi, Hhip*, and *Aspn* (Fig. 4b). However, although detected in cluster 1, transcript levels for these genes were relatively low. This, along with the lack of expression of other marker genes (Supplementary Fig. 8), did not provide us with sufficient confidence to classify the cells of this cluster as either conventional myofibroblasts or pericytes. Nonetheless, given that these retinoid-containing cells were isolated together with conventional lipofibroblast and share lipofibroblast-specific transcriptional patterns, as well as expressing genes required for lipid metabolism such as *Lpl, Lrp1, Lrat, Rbp1*, we consider these cells to be lipofibroblasts committed to the lipogenic-to-myogenic switch[64,87,88] and annotate this cluster as "myofibroblast-committed" lipofibroblasts.

Differentially expressed gene (DEG) analysis allowed for the identification of a highly distinctive expression pattern for cluster 1 fibroblasts (Fig. 4c). It included a significant number of upregulated genes encoding proteins aimed at blocking proliferation, inducing differentiation, and acquiring functions attributed to extracellular matrix (ECM) homeostasis and lipid metabolism. These included transcripts for negative regulators of Tgfβ signaling (*Pmepa1, Cav1, Nr4a1*), transcription factors involved in differentiation (*Klf2, Klf4*), binding proteins with growth-promoting activity (*Igfbp4, Igfbp6*) and proteins enabling growth arrest by either delaying the G1 phase and entry into the S phase (*Sfrp1, Nbl1, Gas1, Gas6*) or G2 phase arrest (*Cdkn1a*). A significant fraction of the top differentially expressed genes in cluster 1 fibroblasts included genes encoding ECM proteins that either define its structure (*Col1a1, Col3a1, Col14a1, Eln, Fbln1, Dpt, Dcn, Mfap5*) or enable ECM remodeling (*Mmp2, Mmp3, Timp1, Cpxm1, Man2a1, Cygb, Mt1, Mt2, Fxyd5, Fxyd6*). Importantly, the high expression of several genes associated with lipid metabolism (*Rbp4, Pltp, C3, C4b*) also was identified for cluster 1 fibroblasts, supporting the idea of their close relation to lipofibroblasts (Fig. 4c).

Taken together, our data allowed us to extend understanding of cellular aspects of retinoid storage in the lung by identifying specific cell populations involved in retinoid accumulation. In addition, these data establish that lung stromal, endothelial, epithelial, and myeloid cells contain intracellular retinoid stores, with transcriptionally distinct subpopulations of pulmonary lipofibroblasts being quantitatively the predominant cellular storage site for lung retinoids. It remains to be understood why each of these different pulmonary cell types takes up and accumulates retinoids, and how this contributes to the specific functions of these cells.

## Lack of local pulmonary retinoids results in amplification of inflammatory cell infiltration, compromised integrity of alveoli, and surfactant production

To gain insight into the events specifically involving retinoids and retinoid-related parameters in the prevention and/or development of ALI, we analyzed the initial events preceding the lethality observed for *Lrat*$^{-/-}$ mice. We focused on responses at an early time point (24 h after instillation) following instillation of a lower, 5 mg/kg LPS dose than the 25 mg/kg dose we used in the groups described above. Experiments involving analyses of intact lung physiology for *Lrat*$^{-/-}$ and matched wild-type (*Lrat*$^{+/+}$) mice were undertaken.

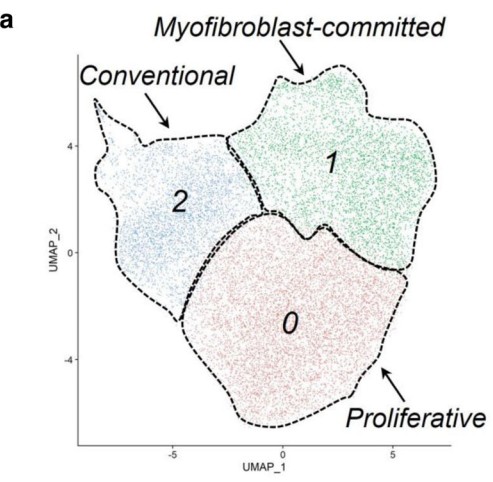

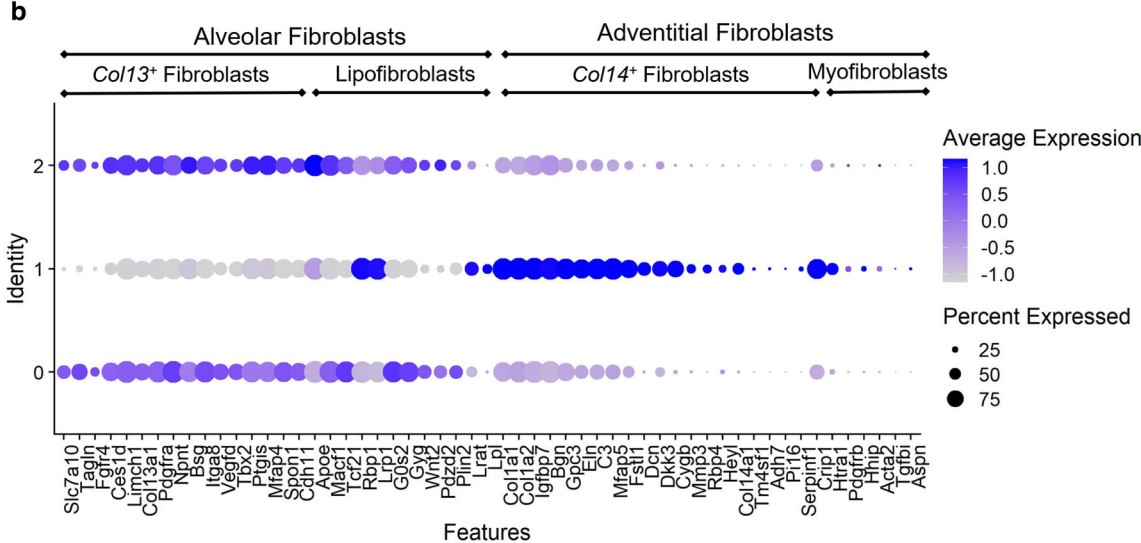

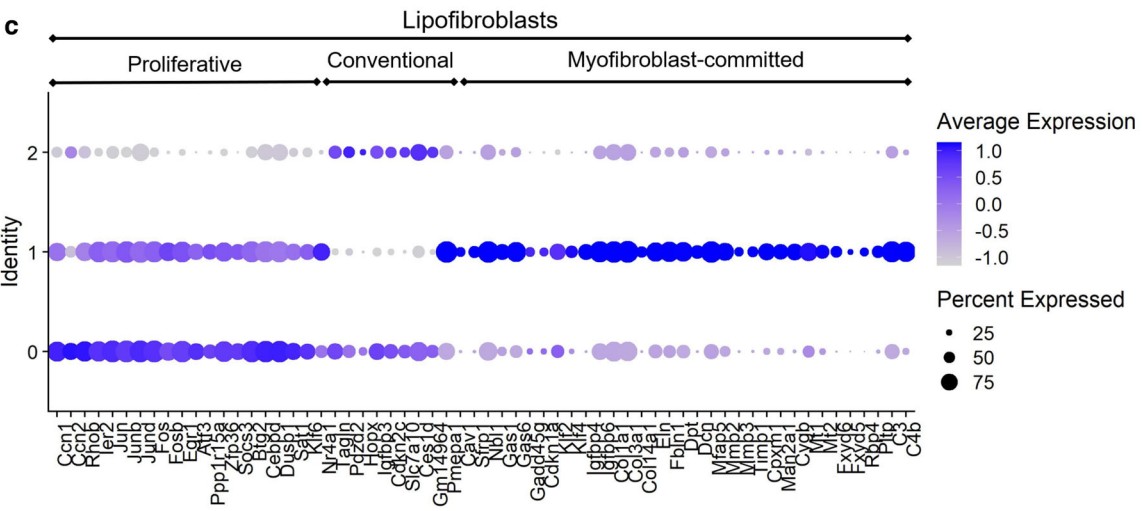

Twenty-four hours after 5 mg/kg LPS instillation, $Lrat^{-/-}$ mice experienced significantly greater inflammatory lung injury than control, as evidenced by significantly higher BAL cell numbers (Fig. 5a), and higher BAL and plasma Tnfα concentrations (Fig. 5b, c). A small but statistically significant ($p = 0.0014$) increase in extravascular lung water (EVLW) was also observed (Fig. 5d). In addition to these changes in $Lrat^{-/-}$ lungs, the lack of pulmonary retinoid stores significantly impacted parameters that define lung functionality. Twenty-four hours after LPS instillation (5 mg/kg), a significant downregulation of mRNA expression of surfactant protein B ($Sftpb$) was detected (Fig. 5e) for both $Lrat^{-/-}$ and control mice. However, the extent of downregulation observed for $Lrat^{-/-}$ mice was significantly greater than that observed for littermate controls. While $Sftpc$ mRNA expression in the LPS-treated $Lrat^{-/-}$ lungs was not different for $Lrat^{-/-}$ mice instilled with the

**Fig. 4 | Postprocessing analysis of single-cell RNA-seq datasets of mouse lung stromal retinoid-containing cells. a** UMAP visualization of stromal cell clustering inferred from mouse lung retinoid-containing cell scRNA-seq datasets; Three retinoid-containing lipofibroblast clusters (proliferative, conventional, and myofibroblast-committed lipofibroblasts) are annotated. **b** Dot plot visualization of marker gene expression of lung fibroblasts (alveolar and adventitial, *Col13*[+] and *Col14*[+], lipofibroblasts, and myofibroblasts) in stromal clusters from retinoid-containing cell scRNA-seq datasets. The *x*-axis (Features) gives gene names, the *y*-axis (Identity) indicates cluster numbers. The level of expression is specified by the color legend. Size of the cell population expressing the gene of interest is indicated by the size of the circle as specified by the legend. **c** Dot plot visualization of differentially expressed genes (DEGs) in stromal clusters (proliferative, conventional, and myofibroblast-committed lipofibroblasts) from retinoid-containing cell scRNA-seq datasets. The *x*-axis (Features) indicates gene names, the y-axis (Identity) indicates cluster numbers. The intensity of expression is indicated by the color legend. Size of the cell population expressing the gene of interest is indicated by the size of the circle as specified in **b** and **c**.

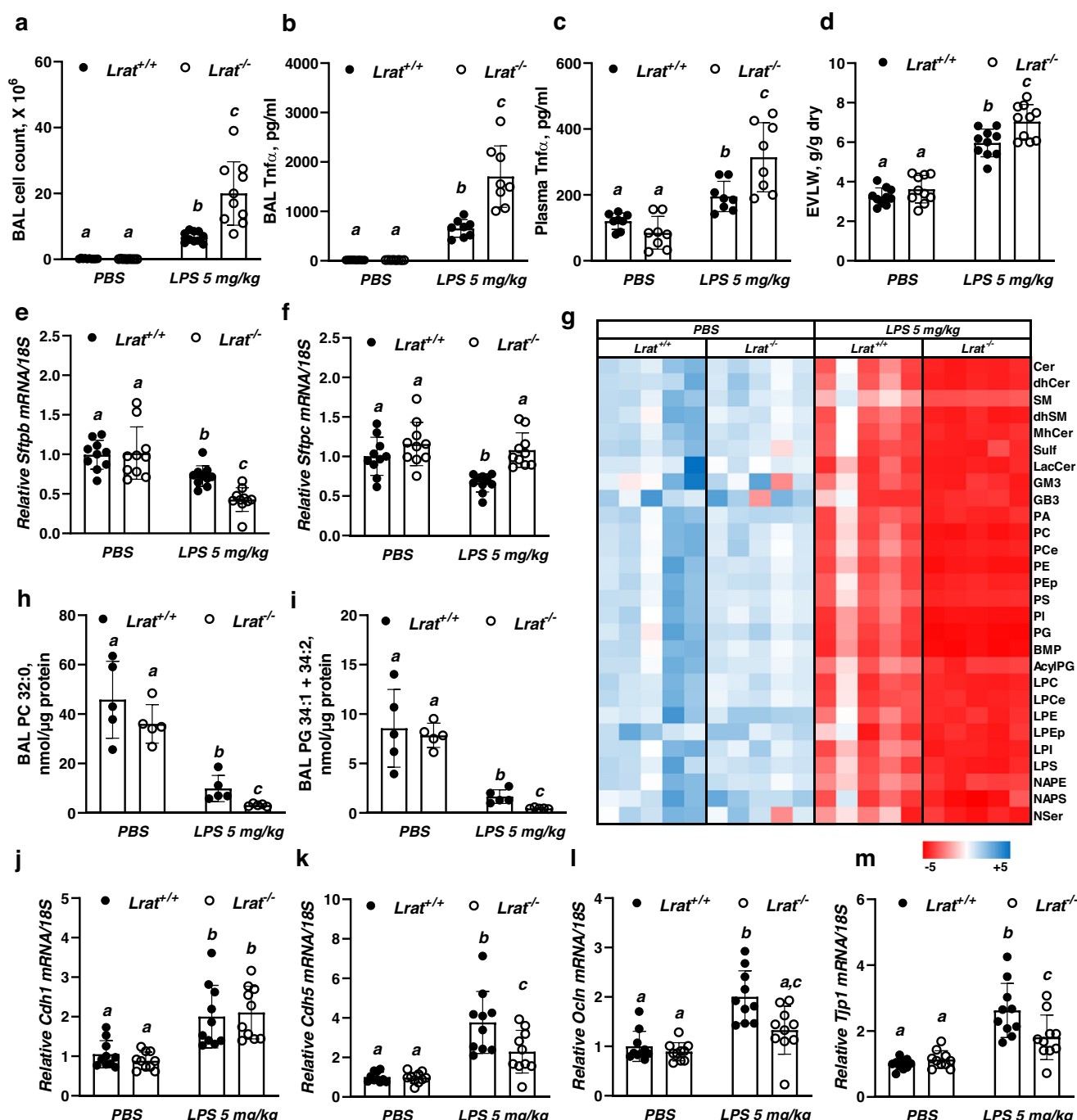

PBS vehicle (Fig. 5f), *Sftpb* transcript levels in the lungs of the *Lrat*[−/−] mice were lower compared to the LPS instilled littermate *Lrat*[+/+] mice (Fig. 5e). Furthermore, a dysregulation in surfactant protein gene expression in the lungs of mice with no retinoid stores was associated with alterations in surfactant lipid composition (Fig. 5g). Lipidomic

profiling of the BAL fluid allowed for the identification of different surfactant-associated lipid species (Fig. 5g, Supplementary Fig. 9, Source Data file). The majority of the BAL fluid lipidome included phosphatidylcholines (PC), phosphatidylglycerols (PG), and dihydro-sphingomyelins (dhSM), comprising ~65, 10, and 10% of the total molar

**Fig. 5 | Lung functional parameters in mice during LPS-induced inflammatory stress.** *a* Bronchoalveolar lavage (BAL) fluid cell numbers in mice 24 h after intranasal instillation of LPS (5 mg/kg of body weight in PBS) or PBS alone. Values marked with different letters (a, b, c) are statistically different (a is different from b, $p = 0.0025$; a is different from c, $p = 6.23e-12$; b is different from c, $p = 3.63e-08$). Statistical differences were first analyzed by a one-way ANOVA followed by multiple comparisons employing Tukey's HSD post hoc test. All values are given as the mean ± 1 S.D., $n = 10$ for each $Lrat^{+/+}$ group and $n = 9$ for each $Lrat^{-/-}$ group. **b** Tnfα concentrations determined by ELISA in cell-free BAL fluid of mice 24 h after intranasal instillation of LPS (5 mg/kg of body weight) or PBS. Values marked with different letters (a, b, c) are statistically different (a is different from b, $p = 0.0005$; b is different from c, $p = 3.63e-11$; b is different from c, $p = 4.81e-07$). Statistical differences were first analyzed by a one-way ANOVA followed by multiple comparisons employing Tukey's HSD post hoc test. All values are given as the mean ± 1 S.D., $n = 8$ for each group. **c** Tnfα concentrations determined by ELISA in plasma of mice 24 h after intranasal instillation of LPS (5 mg/kg of body weight) or PBS. Values marked with different letters (a, b, c) are statistically different (a is different from b, $p = 0.025$; a is different from c, $p = 1.42e-06$; b is different from c, $p = 0.0009$). Statistical differences were first analyzed by a one-way ANOVA followed by multiple comparisons employing Tukey's HSD post hoc test. All values are given as the mean ± 1 S.D., $n = 8$ for each group. **d** Extravascular lung water (EVLW) for mice 24 h after intranasal instillation of LPS (5 mg/kg of body weight) or PBS. Values marked with different letters (a, b, c) are statistically different (a is different from b, $p = 1.82e-10$; a is different from c, $p = 2.13e-14$; b is different from c, $p = 0.0014$). Statistical differences were first analyzed by a one-way ANOVA followed by multiple comparisons employing Tukey's HSD post hoc test. All values are given as the mean ± 1 S.D., $n = 10$ for each group. **e** *W*hole lung *Sftpb* mRNA expression determined by qRT-PCR in mice 24 h after intranasal instillation of LPS (5 mg/kg of body weight in PBS) or PBS alone. Values marked with different letters (a, b, c) are statistically different (a is different from b, $p = 0.0077$; a is different from c, $p = 9.15e-07$; b is different from c, $p = 0.0039$). Statistical differences were first analyzed by a one-way ANOVA followed by multiple comparisons employing Tukey's HSD post hoc test. All values are given as the mean ± 1 S.D., $n = 10$ for each group. **f** Whole lung *Sftpc* mRNA expression determined by qRT-PCR in mice 24 h after intranasal instillation of LPS (5 mg/kg of body weight in PBS) or PBS alone. Values marked with different letters (a, b) are statistically different (a is different from b, $p = 0.0015$). Statistical differences were first analyzed by a one-way ANOVA followed by multiple comparisons employing Tukey's HSD post hoc test. All values are given as the mean ± 1 S.D., $n = 10$ for each group. **g** Heatmap indicating relative concentrations of BAL lipids determined by UPLC-MS/MS in mice 24 h after intranasal instillation of LPS (5 mg/kg of body weight in PBS) or PBS alone. **h** BAL concentrations of dipalmitoylphosphatidylcholine (PC 32:0) determined by UPLC-MS/MS in mice 24 h after intranasal instillation of LPS (5 mg/kg of body weight in PBS) or PBS alone. Values marked with different letters (a, b, c) are statistically different (a is different from b, $p = 1.21e-05$; a is different from c, $p = 1.43e-06$; b is different from c, $p = 0.023$). Statistical differences were first analyzed by a one-way ANOVA followed by multiple comparisons employing Tukey's HSD post hoc test. All values are given as the mean ± 1 S.D., $n = 5$ for each group. **i** BAL concentrations of phosphatidylglycerol (PG 34:1 + PG 34:2) determined by UPLC-MS/MS in mice 24 h after intranasal instillation of LPS (5 mg/kg of body weight in PBS) or PBS alone. Values marked with different letters (a, b, c) are statistically different (a is different from b, $p = 8.21e-05$; a is different from c, $p = 1.43e-05$; b is different from c, $p = 0.0048$). Statistical differences were first analyzed by a one-way ANOVA followed by multiple comparisons employing Tukey's HSD post hoc test. All values are given as the mean ± 1 S.D., $n = 5$ for each group. **j** Whole lung *Cdh1* mRNA expression determined by qRT-PCR in mice 24 h after intranasal instillation of LPS (5 mg/kg of body weight in PBS) or PBS alone. Values marked with different letters (a, b) are statistically different (a is different from b, $p = 0.0005$). Statistical differences were first analyzed by a one-way ANOVA followed by multiple comparisons employing Tukey's HSD post hoc test. All values are given as the mean ± 1 S.D., $n = 10$ for each group. **k** Whole lung *Cdh5* mRNA expression determined by qRT-PCR in mice 24 h after intranasal instillation of LPS (5 mg/kg of body weight in PBS) or PBS alone. Values marked with different letters (a, b, c) are statistically different (a is different from b, $p = 2.08e-07$; a is different from c, $p = 0.0052$; b is different from c, $p = 0.0016$). Statistical differences were first analyzed by a one-way ANOVA followed by multiple comparisons employing Tukey's HSD post hoc test. All values are given as the mean ± 1 S.D., $n = 10$ for each group. **l** Whole lung *Ocln* mRNA expression determined by qRT-PCR in mice 24 h after intranasal instillation of LPS (5 mg/kg of body weight in PBS) or PBS alone. Values marked with different letters (a, b, c) are statistically different (a is different from b, $p = 2.45e-06$; b is different from c, $p = 0.0006$). Statistical differences were first analyzed by a one-way ANOVA followed by multiple comparisons employing Tukey's HSD post hoc test. All values are given as the mean ± 1 S.D., $n = 10$ for each group. **m** Whole lung *Tjp1* mRNA expression determined by qRT-PCR in mice 24 h after intranasal instillation of LPS (5 mg/kg of body weight in PBS) or PBS alone. Values marked with different letters (a, b, c) are statistically different (a is different from b, $p = 1.25e-07$; a is different from c, $p = 0.0023$; b is different from c, $p = 0.0023$). Statistical differences were first analyzed by a one-way ANOVA followed by multiple comparisons employing Tukey's HSD post hoc test. All values are given as the mean ± 1 S.D., $n = 10$ for each group. Values marked with different letters (a, b, c) are statistically different (a is different from b, $p = 2.45e-06$; b is different from c, $p = 0.0006$). Statistical differences were first analyzed by a one-way ANOVA followed by multiple comparisons employing Tukey's HSD post hoc test. All values are given as the mean ± 1 S.D., $n = 10$ for each group.

lipid content respectively (Supplementary Fig. 9). Other detectable lipid species included phosphatidylethanolamines (PE), phosphatidylinositols (PI), phosphatidylcholine ethers (PCe), plasmalogen phosphatidylethanolamines (Pep), sphingomyelins (SM), bis(monoacylglycero)phosphates (BMP), phosphatidylserine (PS), lysophosphatidylcholine (LPC), and monohexosylceramides (MhCer), comprising ~15% of the total molar lipid content when combined, with percentage for each lipid species ranging from 0.3 to 3%. The fraction of other lipid species was <1% (see a complete list with mol % in Supplementary Fig. 9). This analysis revealed significant alterations in major surfactant-associated lipid species, especially phosphatidylcholines (PC) and phosphatidylglycerols (PG), whose concentrations were significantly lower in LPS-instilled $Lrat^{-/-}$ mice (Fig. 5h, i).

mRNA expression of genes encoding junction proteins needed for maintaining integrity of the air-blood barrier, including *Cdh5*, *Ocln*, and *Tjp1* was downregulated in $Lrat^{-/-}$ lungs 24 h after LPS instillation compared to littermate $Lrat^{+/+}$ controls (Fig. 5j–m).

Our assessments of these functional parameters provide experimental evidence for the development of more severe ALI when local retinoid storage is absent or diminished. Taken together with the survival data (Fig. 1e), these data establish that insufficiency of pulmonary REs stores exacerbates ALI by causing deficits in alveolar integrity, surfactant production and by increasing lung inflammation.

## The pulmonary retinoid metabolome and transcriptome undergo extensive changes in response to ALI

To identify possible crosstalk between alterations in lung functions and retinoid physiology, we explored how retinoid metabolism changes in the context of ALI development in the intact lung. Consistent with our observations from experiments employing the 25 mg/kg LPS dose (Fig. 1a), even LPS instillation at the dose of 5 mg/kg to mice resulted in an ~50% decline in total lung REs within 24 h (Fig. 6a). This was associated with elevation in lung ROH (Fig. 6b), suggesting increased hydrolysis of these pulmonary RE stores during the acute inflammatory response. ROH concentrations in the $Lrat^{-/-}$ lungs (with no detectable REs) significantly declined (Fig. 6b). While plasma ROH concentrations declined almost 50% in wild-type ($Lrat^{+/+}$) mice, they remained unchanged in $Lrat^{-/-}$ animals 24 h after 5 mg/kg LPS instillation (Fig. 6c). Most importantly, a >2-fold elevation in pulmonary ATRA concentrations was observed 24 h after LPS (5 mg/kg) instillation for wild-type ($Lrat^{+/+}$) mice (Fig. 6d). These data lead us to propose that the early period of ALI is associated with extensive local pulmonary retinoid mobilization aimed at providing local substrate needed for ensuring ATRA synthesis, which is required for maintaining RAR-dependent transcription. Indeed, qRT-PCR analysis of transcripts for pulmonary genes indicated that those involved in retinoid metabolism and signaling were highly responsive to inflammatory insult within the first 24 h (Fig. 6e–i). Among the highly upregulated genes were ones

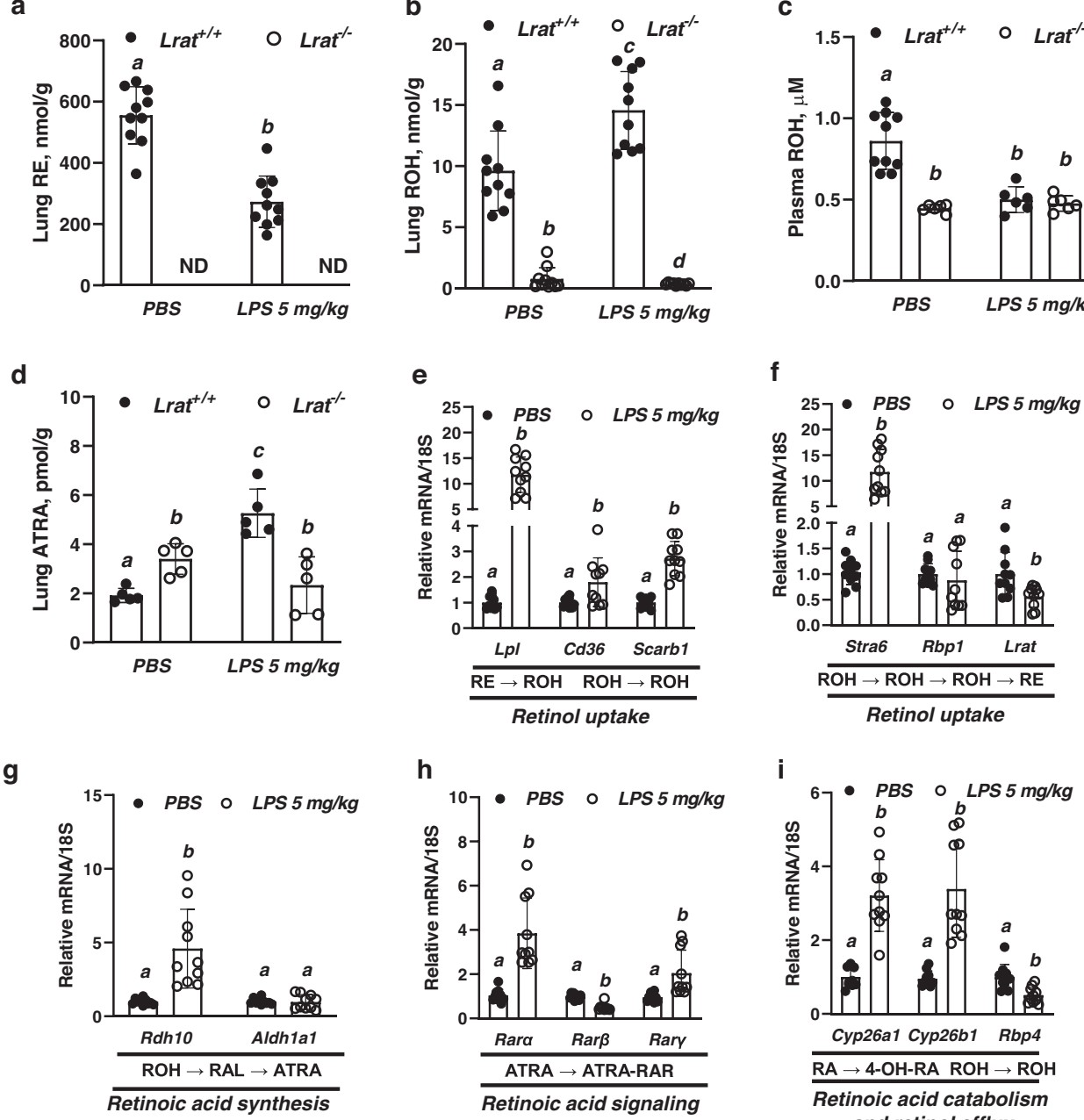

**Fig. 6 | Retinoid concentrations and retinoid-related transcript levels of genes mediating retinoid metabolism in mouse lungs during early periods of ALI.**
**a** Lung total retinyl ester (RE, nmol/g) concentrations determined by HPLC in mice 24 h after intranasal instillation of LPS (5 mg/kg of body weight in PBS) or PBS alone. Values marked with different letters (a, b) are statistically different (a is different from b, $p = 1.28\text{e-}06$). Statistical differences were analyzed by a one-way ANOVA. All values are given as the mean ± 1 S.D., $n = 10$ for each group. **b** Lung retinol (ROH, nmol/g) concentrations determined by HPLC in mice 24 h after intranasal instillation of LPS (5 mg/kg of body weight in PBS) or PBS alone. Values marked with different letters (a, b, c, d) are statistically different (a is different from b, $p = 3.63\text{e-}10$; c is different from d, $p = 7.50\text{e-}16$; a is different from c, $p = 3.11\text{e-}05$). Statistical differences were first analyzed by a one-way ANOVA followed by multiple comparisons employing Tukey's HSD post hoc test. All values are given as the mean ± 1 S.D., $n = 10$ for each group. **c** Plasma retinol (ROH, μM) concentrations determined by HPLC in mice 24 h after intranasal instillation of LPS (5 mg/kg of body weight in PBS) or PBS alone. Values marked with different letters (a, b) are statistically different (a is different from b, $p = 2.95\text{e-}06$). Statistical differences were first analyzed by a one-way ANOVA followed by multiple comparisons employing Tukey's HSD post hoc test. All values are given as the mean ± 1 S.D.,

$n = 10$ for PBS instilled $Lrat^{+/+}$ group, $n = 6$ for all other groups. **d** All-*trans*-retinoic acid concentrations (ATRA) (pmol/g) determined by UPLS-MS/MS in mice 24 h after intranasal instillation of LPS (5 mg/kg of body weight in PBS) or PBS alone. Values marked with different letters (a, b, c) are statistically different (a is different from b, $p = 0.0123$; a is different from c, $p = 9.39\text{e-}06$; b is different from c, $p = 4.22\text{e-}05$). Statistical differences were first analyzed by a one-way ANOVA followed by multiple comparisons employing Tukey's HSD post hoc test. All values are given as the mean ± 1 S.D., $n = 5$ for each group. **e–i** Whole lung mRNA expression of genes mediating retinol uptake (*Lpl*, *Cd36*, *Scarb1*, *Stra6*, *Rbp1*, *Lrat*), all-*trans*-retinoic acid synthesis (*Rdh10*, *Aldh1a1*), signaling (*Rara*, *Rarβ*, *Rary*), catabolism (*Cyp26a1*, *Cyp26b1*) and efflux (*Rbp4*) in mice 24 h after intranasal instillation of LPS (5 mg/kg of body weight in PBS) or PBS alone. Values marked with different letters (a, b) are statistically different (a is different from b, $p = 1.21\text{e-}08$ (for *Lpl*), $p = 0.018$ (for *Cd36*), $p = 3.66\text{e-}07$ (for *Scarbp1*), $p = 4.76\text{e-}07$ (for *Stra6*), $p = 0.0092$ (for *Lrat*), $p = 0.0005$ (for *Rdh10*), $p = 3.25\text{e-}05$ (for *Rara*), $p = 4.12\text{e-}07$ (for *Rarβ*), $p = 0.0045$ (for *Rary*), $p = 1.84\text{e-}06$ (for *Cyp26a1*), $p = 1.74\text{e-}05$ (for *Cyp26b1*), $p = 0.0012$ (for *Rbp4*)). Statistical differences were analyzed by a one-way ANOVA. All values are given as the mean ± 1 S.D., $n = 10$ for each group.

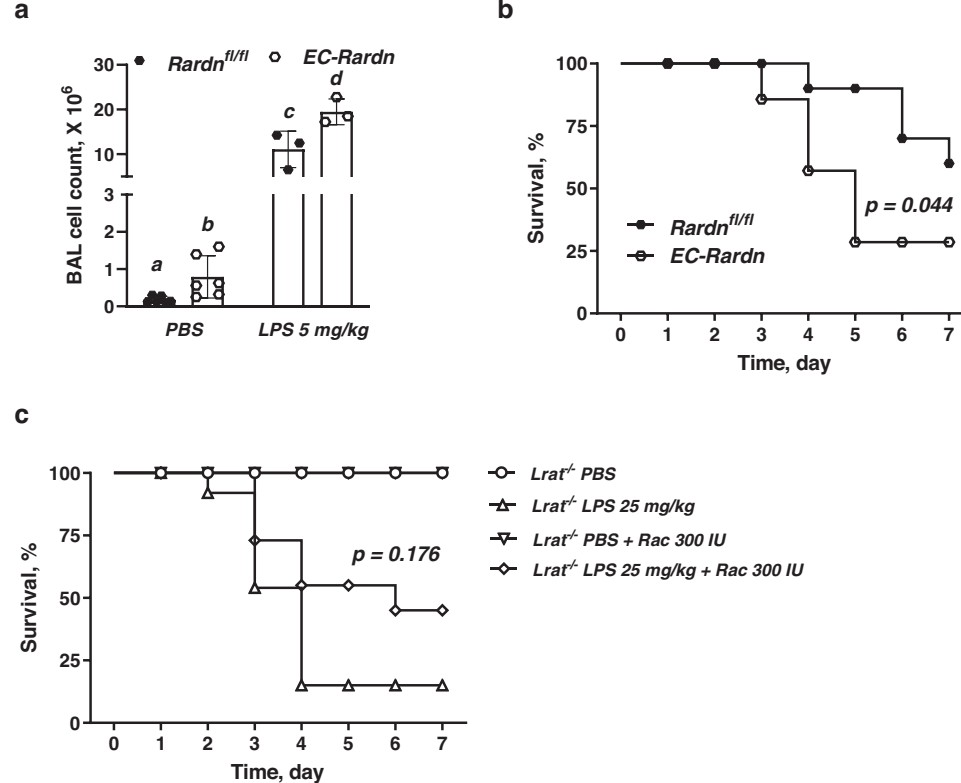

**Fig. 7 | Loss- and gain-of-function mouse models for lung cell retinoid signaling during LPS-induced ALI. a** Bronchoalveolar lavage (BAL) fluid cell numbers in EC-Rardn mice 24 h after intranasal instillation of LPS (5 mg/kg of body weight in PBS) or PBS alone. Values marked with different letters (a, b, c, d) are statistically different (a is different from b, $p = 0.026$; c is different from d, $p = 0.043$). Statistical differences were analyzed by a one-way ANOVA. All values are given as the mean ± 1 S.D., $n = 6$ for PBS group, $n = 3$ for LPS group. **b** Kaplan–Meier survival curves for EC-Rardn mice as well as for their respective littermate controls (*Rardn*$^{fl/fl}$ mice) over the 7-day period after intranasal instillation of a LPS (25 mg/kg of body weight in PBS) or PBS alone. Statistically significant differences ($P = 0.044$) were identified using the log rank test to compare survival curves for LPS-instilled EC-Rardn and *Rardn*$^{fl/fl}$ mice, $n = 8$ for each group. **c** Kaplan–Meier survival curves for *Lrat*$^{-/-}$ mice supplemented with 300 IU of retinol over the 7-day period after intranasal instillation of a LPS (25 mg/kg of body weight in PBS) or PBS alone, $n = 14$ for each group. $P = 0.176$ denotes the lack of statistically significant difference calculated using the log rank test comparing the survival curves for LPS-instilled *Lrat*$^{+/+}$ and retinoid-supplemented *Lrat*$^{-/-}$ mice, $n = 14$ for each group.

needed to either facilitate the uptake of retinol, a substrate for ATRA synthesis, or genes encoding proteins enabling ATRA synthesis and signaling. For instance, the expression of genes needed to facilitate retinol uptake from lipoproteins (*Lpl*, *Cd36*, *Scarb1*) were all significantly upregulated in the lungs of LPS-instilled mice (Fig. 6e). Surprisingly, *Stra6* mRNA expression, which was nearly undetectable in the normal lung, was upregulated >10-fold in the lungs during ALI, presumably as an additional route for acquiring needed retinoid (Fig. 6f). qRT-PCR analysis of *Stra6* expression in cells collected from BAL fluids suggests that the observed upregulation in *Stra6* expression in the whole lung tissue mRNA preparations was primarily associated with inflammatory *Tnfα*-expressing cells that infiltrate lung during LPS-induced inflammation (Supplementary Fig. 10). Furthermore, about a 5-fold upregulation was seen for pulmonary mRNA expression of *Rdh10*, a gene that encodes the first enzyme needed for ATRA synthesis from ROH[89] (Fig. 6g). In addition, transcript levels for *Rara* and *Rarγ*, genes encoding nuclear proteins that mediate ATRA-dependent transcriptional regulation, were elevated in the mouse lungs 24 h after LPS instillation (Fig. 6h). However, transcription of *Rarβ*, the least expressed Rar isoform in the lung, was downregulated. Concurrently, expression of *Lrat*, which could negatively affect ROH availability for ATRA production by converting it into RE, as well as *Rbp4*, which could contribute to ROH efflux from the lungs thus preventing ATRA synthesis, were downregulated (Fig. 6i). However, mRNA levels of some genes mediating retinoid metabolism in the lung remained unchanged for PBS and LPS instilled animals, although showing greater

variability in LPS instilled mice. These included the gene for *Rbp1* as well as *Aldh1a1*, which encodes the second and final enzyme needed for ATRA synthesis (Fig. 6f, g). Importantly, mRNA expression of *Cyp26a1* and *Cyp26b1*, two retinoic acid-responsive monooxygenases that catalyze the first step of ATRA catabolic inactivation[49], were upregulated reflecting high intracellular ATRA concentrations. Since *Cyp26a1* and *Cyp26b1* upregulation is aimed at buffering ATRA excess, this suggests that there is dysregulation of pulmonary retinoid homeostasis during the early stages of ALI.

Our data establish that early in the lung's responses to experimentally induced ALI there is an extensive mobilization of local pulmonary REs aimed at providing ROH for enzymatic conversion to transcriptionally active ATRA. However, unlike in wild-type (*Lrat*$^{+/+}$) animals, pulmonary concentrations of ATRA in *Lrat*$^{-/-}$ lungs remain low (Fig. 6d) due to the lack of sufficient local retinoid stores that are needed to support rapid ATRA synthesis for sustaining normal lung functions. These data lead us to conclude that the absence of retinoid stores in the lungs of *Lrat*$^{-/-}$ mice results in diminished levels of ATRA and an impaired ability of the lung to maintain retinoid-dependent transcription responses in the face of an inflammatory stimulus. This then more quickly leads to impaired surfactant lipid homeostasis, which requires ATRA for pulmonary phospholipid biosynthesis[90], diminished transcription of *Sfptb*, a known retinoid-responsive gene encoding surfactant protein B[20,21], as well as other genes encoding junction proteins that are needed for maintaining

alveolar integrity, including *Cdh5*, *Ocln*, and *Tjp1*. Hence, our data indicate that these molecular events, if not corrected, will give rise to a more adverse outcome.

### RAR-mediated ATRA signaling in ECs is critical for survival during ALI

To address the importance of cell-specific ATRA signaling for the survival during ALI, a mouse model with ablated Rar signaling in ECs (EC-Rardn mice) was generated using an approach and mice we previously employed in our lab in other published studies[91,92]. To generate EC-Rardn mice, a mouse line harboring a transgene encoding dominant/negative retinoic acid receptor-α403 (Rardn) mutant protein (*Rardn*[fl/fl] mice[91,92]) was crossed with mice expressing Cre-recombinase (Cre) under the control of a tamoxifen-inducible EC-specific vascular endothelial cadherin (*VE-Cad/Cdh5*) promotor (EC-CreER[T2] mice[93]). Rardn expression suppresses transcription of all three Rars, thus leading to the near-complete ablation of ATRA signaling[94,95]. One month after the final tamoxifen injection aimed at activating the transgene, cell-specific Rardn expression was confirmed by immunoblot of the c-*myc*-epitope tag of the Rardn construct in primary endothelial cells isolated from EC-Rardn mice (Supplementary Fig. 11).

Rardn overexpression in ECs and ablation of Rar-mediated ATRA signaling were associated, one month after tamoxifen dosing, with increased inflammatory cell infiltration of the lungs. This was detected even without LPS administration (Fig. 7a). Administration of 5 mg/kg of LPS resulted in a more pronounced inflammatory response in the lungs of EC-Rardn mice as evidenced by significantly higher BAL cell counts 24 h after the LPS administration (Fig. 7a). Most importantly, the survival rate of EC-Rardn mice over a 7-day period after 25 mg/kg LPS instillation was significantly lower (only 30%) compared to matched *Rardn*[fl/fl] littermates which had a survival rate of >60% (Fig. 7b). There was a statistically significant difference ($p < 0.044$) between the survival curves for LPS-treated EC-Rardn and *Rardn*[fl/fl] mice as calculated by the Kaplan–Meier log rank test. This recapitulates the greater lethality phenotype we earlier observed for *Lrat*[−/−] mice (Fig. 1e).

### Dietary retinoids can rescue ALI associated lethality in *Lrat*[−/−] mice

A gain-of-function study was carried out to establish the extent to which lipoprotein-derived retinoids are important for maintaining or improving lung health. LPS (25 mg/kg) instilled *Lrat*[−/−] mice were given by gavage a daily dose of 300 IU of retinol over a 7-day study period. In *Lrat*[−/−] mice, supplementation resulted in reduced lethality by 30%, bringing the survival rate to almost 50% (as compared to 15% in LPS instilled *Lrat*[−/−] mice not receiving the oral retinoid) (Fig. 7c). Importantly, based on a Kaplan–Meier log rank test, we did not observe a statistically significant difference ($p = 0.176$) between the survival curves for LPS-treated *Lrat*[+/+] and *Lrat*[−/−] mice receiving retinol. Because dietary retinol absorbed in the intestine is incorporated into chylomicrons, the beneficial effects of retinol supplementation for rescuing lethality support the idea that lipoprotein-derived retinoids taken up by lung cells have an important role in supplying retinoids for maintaining or improving lung health.

## Discussion

Retinoid metabolism is complex and ultimately geared towards forming and maintaining cellular concentrations of ATRA within a very tight range (Supplementary Fig. 1a). ATRA is always found within cells and tissues at very low levels, in the nM range, and is enzymatically formed from much more abundant precursors, ROH and RE, which are usually present within cells and tissues in the μM range. This metabolism involves the actions of many unique enzymes (like Lrat and Rdh10) and unique intracellular and extracellular binding proteins (like Rbp1 and Rbp4), but it also shares some metabolic processes that are utilized by other more abundant neutral lipids (like Lpl, Cd36 and

chylomicrons). It has long been thought that ROH is mobilized from liver stores and delivered to cells by Rbp4 where ROH is taken up and converted to ATRA for direct use within the cell where it is formed to regulate transcription. Our findings indicate that this understanding of retinoid physiology is incorrect for the lung. And, we suspect, this misunderstanding has resulted in incorrect interpretations of findings reported in the literature. We[75,96,97] and others[98,99] have shown that the postprandial uptake of dietary retinoids is a major pathway used by some cells and tissues within the body (i.e. adipose tissue) for accumulating retinoid that can be used for maintaining normal cellular functions. This is in addition to, or as an alternative to, Rbp4 mediated delivery. The data we have provided strongly support the conclusions that postprandial delivery is responsible for allowing lung lipofibroblasts, endothelial cells and alveolar epithelial cells to accumulate retinoid stores and that these local cellular stores within the alveolus are needed for maintaining optimal lung health. These findings are paradigm shifting. Rather than one particular cell type, the lipofibroblast, being the sole cellular site for retinoid storage within the alveolus, a number of different cell types, ones in close anatomic proximity, have the metabolic capability for accumulating retinoid stores and mobilizing them. This finding was unanticipated before the present study. Moreover, it changes how retinoid biology upstream of ATRA actions in transcription is understood.

The ability of lungs to accumulate retinoids locally has long been attributed to the pulmonary lipofibroblasts, lipid-laden interstitial cells located in the alveolar niche between alveolar capillary endothelial cells and the alveolar epithelial cells present in both murine[32,63,64,100] and human lungs[83,101,102] (Supplementary Fig. 1b). Using single-cell transcriptomic approaches to characterize pulmonary cells that possess endogenous retinoid fluorescence, we unexpectedly observed cellular diversity in retinoid storage within the lungs. An extensive body of literature underscores the importance of lung lipofibroblasts as well as lipofibroblast-derived retinoids for alveoli formation in pre- and postnatal lung development[63,64,100,103–106]. Our current study extends this knowledge to adult lung physiology during ALI. Our study provides experimental evidence regarding the transcriptional diversity of the retinoid-containing lipofibroblasts within the lungs. Unique transcriptional signatures allowed us to identify three distinct lipofibroblast clusters (Fig. 4). However, proper annotation of these fibroblast clusters is a challenging task given the heterogeneity of pulmonary fibroblasts and the lack of consensus in the rapidly evolving scRNA-seq literature regarding which transcriptional markers should be used for proper cell annotation[82–86]. Owing to the different approaches used for cell isolations and data analyses, the same set of expression markers have been used by different investigators to annotate different lung cell types[82–86]. Moreover, markers that were identified and proposed for annotation by some investigators cannot be reproduced by others[82–86]. To overcome any potential discrepancy in our cell annotations and to be consistent with the literature, we have analyzed our scRNA-seq integrated datasets using only markers for which there is consensus in the literature (Fig. 4b, Supplementary Fig. 8)[82–86]. We consider all of the fibroblasts from our dataset to be lipofibroblasts due to the presence of endogenous retinoid fluorescence and with their unique transcriptional signatures; these include "conventional" lipofibroblasts, "proliferative" lipofibroblasts, and "myofibroblast-committed" lipofibroblasts (Fig. 4). Our data further establish that genes involved in mediating retinoid and lipid metabolism can be highly discriminatory markers for annotating these diverse lipofibroblast populations. For instance, *Rbp4*, a gene encoding a protein involved in ROH extracellular transport, was expressed solely by a cluster of "myofibroblast-committed" lipofibroblasts (Fig. 4). *Rbp4* was amongst the genes that exhibit the highest cell-to-cell variation in the whole dataset (Supplementary Fig. 8f). However, the function of Rbp4 within lipofibroblasts and more generally in alveolar physiology remains to be identified.

An important finding from this study is that other non-fibroblast cell types co-isolated during sorting based on retinoid autofluorescence (Fig. 3, Supplementary Figs. 6 and 7). Certain subpopulations of pulmonary endothelial, epithelial and myeloid cells contain retinyl esters and all express both *Rbp1* and *Lrat* (Fig. 3), genes that are needed for effective ROH uptake and RE synthesis. With regards to *Lrat* expression, only one published study reports its expression in endothelial cells found in normal rodent and human livers[107]. By establishing that lipofibroblasts are not the only cell type capable of retinoid uptake and accumulation in the lung, our data highlight the complexity of cell interactions within the alveolar niche that are essential for mediating retinoid and lipid metabolism. These findings raise new questions regarding the molecular mechanisms underlying retinoid transfer to and uptake by lipofibroblasts. Given the spatial separation of lipofibroblasts from the circulation by the endothelial cell barrier (Supplementary Fig. 1b), these mechanisms will need to be established in future investigations.

A very striking observation from our work is the greater mortality rate observed for LPS-treated *Lrat*[-/-] mice, which lack local lung retinoid stores, compared to that of matched wild-type *Lrat*[+/+] mice or *Rbp4*[-/-] mice, both of which possess local stores. This establishes the importance of local cellular retinoid stores. Our data show that without these local stores, mice experiencing ALI fail more quickly to maintain expression levels of genes needed to assure tight cellular contacts, normal surfactant concentrations and composition, and experience greater fluid and macrophage accumulation within the alveolus. This gives rise to more severe disease and increased mortality. These changes affecting lung health are accompanied by changes in retinoid- and lipid-related gene expression. This suggests that lung retinoid and lipid metabolism are being repurposed towards immediately acquiring more retinoid for use in ATRA synthesis; presumably to allow for restoration of control of critical transcriptional responses. Thus, our data establish that local RE stores, as well as uncompromised retinoid metabolism within the cells composing the alveoli, limits the severity of an LPS-induced inflammatory challenge.

In summary, we have defined the molecular actions of retinoids in the pathophysiology of ALI. Our data underscore the importance of local lung retinoid storage as the metabolic driver during ALI, as opposed to systemic retinoid transport and metabolism, for ensuring maintenance of lung cell-specific functions, including maintenance of alveolar integrity, surfactant production, and inflammatory response.

## Methods

### Animal husbandry and manipulations

All experiments involving mice were carried out with the approval of the Institutional Animal Care and Use Committee of Columbia University according to criteria outlined in the Guide for the Care and Use of Laboratory Animals prepared by the National Academy of Sciences[108] Mice were housed under a 12 h light/dark cycle at ambient temperature between 65–75 °F (-18–23 °C) with 40-60% humidity and with ad libitum access to water and food. Groups of 3-month-old (10–12 weeks-old) male mice were used in all studies. Depending on the experiment, 5–14 animals per each group were employed. Lecithin:retinol acyltransferase-deficient mice (*Lrat*-deficient, *Lrat*[-/-] mice[47]) on a C57BL/6 genetic background and age-matched littermates (wild-type, *Lrat*[+/+] mice) were used in the experiments. Retinol-binding protein 4 deficient mice on a C57BL/6 J/129 SV mixed genetic background (*Rbp4*-deficient, *Rbp4*[-/-] mice[34]) and age-matched littermate controls (wild-type, *Rbp4*[+/+] mice) were used in experiments. For cell-specific ablation of RAR signaling in ECs, a mouse line that expresses a dominant/negative retinoic acid receptor-α403 mutant protein (Rardn) specifically in ECs (EC-Rardn mice) was generated. Mice harboring a floxed *Rardn* transgene (*Rardn*[fl/fl] mice on a mixed genetic background as described in[91,92] were crossed with mice expressing Cre-recombinase (Cre) under the control of a tamoxifen-inducible EC-

specific vascular endothelial cadherin (*VE-Cad/Cdh5*) promotor (EC-CreER[T2] mice) as described in ref. [93]). Cre expression in EC-Rardn mice was induced through ip injection of 2 mg of tamoxifen once every 24 h for a total of 5 consecutive days. Littermate controls (*Rardn*[fl/fl] mice with no Cre gene) were subjected to the same tamoxifen treatment. One month after the final tamoxifen injection, cell-specific Rardn expression in ECs was confirmed by immunoblot of the c-myc-epitope tag of the Rardn construct in primary lung ECs isolated (see below) from EC-Rardn mice (Supplementary Fig. 11).

To induce lung injury and the development of the acute lung injury (ALI), anesthetized mice (intraperitoneal injection of 100 mg/kg ketamine and 5 mg/kg xylazine) from each genotype group were given a dose of lipopolysaccharide (LPS) in sterile phosphate-buffered saline (PBS) via intranasal instillation. LPS concentration was 25 mg/kg body weight (lethal dose) for the ALI survival experiments and 5 mg/kg body weight (nonlethal dose) for early phase ALI experiments. Control animals were instilled with an equal volume of sterile PBS[109]. After the procedures, mice were kept on a warm pad for recovery and were visibly observed and monitored every 15 min during recovery from anesthesia until the animals were fully ambulatory.

In the ALI survival experiments, mice from each of six genotypic groups (*Lrat*[-/-] and *Lrat*[+/+], *Rbp4*[-/-] and *Rbp4*[+/+], EC-Rardn and *Rardn*[fl/fl]) received either 25 mg/kg of LPS dissolved in sterile PBS or PBS alone via a single intranasal instillation followed by monitoring of parameters, including grooming, gait, locomotion, respiration, reflex, body weight changes, and survival over a 7-day period.

In the vitamin A-supplementation experiment, separate groups of either LPS or vehicle (PBS) treated *Lrat*[+/+] and *Lrat*[-/-] mice additionally received 300 IU of retinol in the form of retinyl acetate each day for 7 days. This is ~10-times greater than the amount a mouse acquires daily from consumption of a standard chow diet (30 IU) but is not sufficient to induce either acute or chronic retinoid toxicity. The mice were euthanized 7 days after the instillation of LPS using an approved method of anesthesia, such as $CO_2$ inhalation or ketamine/xylazine injection followed by cervical dislocation and a second method of euthanasia such as thoracotomy or removal of essential organs. At the time of euthanasia, lungs, liver, and blood were collected. The dissected tissues were quickly weighed, frozen in liquid $N_2$, and stored at −80 °C continuously without thaw until analysis.

For the early phase ALI experiments, groups of mice (*Lrat*[+/+] and *Lrat*[-/-], EC-Rardn and *Rardn*[fl/fl]) were euthanized 24 h after 5 mg/kg LPS instillation. Bronchoalveolar (BAL) fluid, lungs, liver, and blood were collected at the time of euthanasia. For BAL fluid collection, the lungs of anesthetized mice (intraperitoneal injection of 100 mg/kg ketamine and 5 mg/kg xylazine) were lavaged 5-times with ice-cold PBS without calcium and magnesium (1 mL each time) through a tracheal cannula. The dissected tissues were quickly weighed, frozen in liquid $N_2$, and stored at -80 °C continuously without thaw until analysis.

### ALI feature assessment

To assess ALI development in our mouse models, several parameters were evaluated according to the American Thoracic Society recommendations[110]. BAL fluids collected from each mouse were centrifuged for 10 min at 350 × g and 4 °C. For cell counts, pellets were resuspended in 1 mL of PBS and stained with Turk's solution. Cell counts were performed using a hemocytometer (Hausser Scientific). The supernatant was analyzed for protein concentration using a Bio-Rad Protein Assay (Bio-Rad Laboratories, Inc.). BAL and plasma Tnfα concentration were assessed using a mouse Tnfα ELISA kit (Enzo Life Sciences, Inc.) according to the manufacturer's protocol. Plasma C-reactive protein concentrations were assessed using a mouse C-reactive protein/CRP ELISA kit (R&D Systems) according to the manufacturer's protocol. For surfactant phospholipid analyses, BAL fluid supernatants were further centrifuged for 30 min at

13,000 × g at 4 °C to pellet cell debris. The resultant supernatants were used for further lipidomic analysis by UPLC/MS/MS (see below for details).

Alveolar capillary barrier integrity was assessed by measuring extravascular lung water (EVLW) content based on the wet-dry ratio of the lung homogenate, corrected for blood water content[109].

## HPLC separation and analyses of retinol and retinyl esters

Retinyl esters and retinol were extracted from plasma, liver, lung, and pulmonary UV-positive cells under a dim yellow safety light and analyzed by HPLC as described earlier[111]. Briefly, livers and lungs were homogenized in 10 vol of PBS (10 mM sodium phosphate, pH 7.2, 150 mM sodium chloride) using a tissue tearer. An aliquot of plasma, (100 μL), tissue homogenate (equivalent to 50 μg of tissue) or cell suspension (200 μL containing 20 to 500 × 10⁶ cells) was then treated with an equal volume of absolute ethanol containing a known amount of retinyl acetate as an internal standard. The retinoids present in either plasma, tissue homogenates, or cell suspensions were extracted into hexane. The hexane extract was dried under a stream of $N_2$ and redissolved in benzene. For determination of tissue and plasma levels of retinol and retinyl esters, a reverse-phase HPLC method employing a 4.6 × 250 mm Symmetry C18 Column (Waters, Milford, MA, USA) and a running solvent consisting of acetonitrile/methanol/methylene chloride (70:15:15 v/v) flowing at 1.8 mL/min. A Waters HPLC system (Waters 717 Autosampler, Waters 515 HPLC Pump, and Waters 2996 Photodiode Array Detector) controlled by Empower version 2 software was employed for these analyses. Retinol and retinyl esters were detected at 325 nm. Quantitation was based on comparisons of the area under the peaks and spectra for unknown samples to those of known amounts of standards. The recovery of internal standard was employed to correct for loss during extraction. For this analysis, HPLC grade solvents were purchased from Thermo Fisher Scientific (Pittsburgh, PA).

## Lipidomic profiling of BAL fluid

Lipidomic profiling of BAL fluid samples was performed using Ultra Performance Liquid Chromatography-Tandem Mass Spectrometry (UPLC-MS/MS) at the Biomarkers Core Laboratory (Irving Institute of Clinical and Translational Research at Columbia University). Lipid extracts were prepared from BAL fluid samples using a modified Bligh and Dyer extraction[112], spiked with appropriate internal standards, and analyzed on a platform comprising Agilent 1260 Infinity HPLC integrated to Agilent 6490 A QQQ mass spectrometer controlled by Masshunter v 7.0 (Agilent Technologies, Santa Clara, CA). Glycerophospholipids and sphingolipids were separated with normal-phase HPLC as described[113]. An Agilent Zorbax Rx-Sil column (2.1 × 100 mm, 1.8 μm) maintained at 25 °C and at a flow rate of 300 μL/min was used under the following conditions: mobile phase A (chloroform:methanol:ammonium hydroxide, 89.9:10:0.1, v/v) and mobile phase B (chloroform:methanol:water:ammonium hydroxide, 55:39:5.9:0.1, v/v); 95% A for 2 min, decreased linearly to 30% A over 18 min and further decreased to 25% A over 3 min, before returning to 95% over 2 min and held for 6 min.

Quantification of lipid species was accomplished using multiple reaction monitoring (MRM) transitions[113–115] under both positive and negative ionization modes in conjunction with referencing of appropriate internal standards: PA 14:0/14:0, PC 14:0/14:0, PE 14:0/14:0, PG 15:0/15:0, PI 17:0/20:4, PS 14:0/14:0, BMP 14:0/14:0, APG 14:0/14:0, LPC 17:0, LPE 14:0, LPI 13:0, Cer d18:1/17:0, SM d18:1/12:0, dhSM d18:0/12:0, GalCer d18:1/12:0, GluCer d18:1/12:0, LacCer d18:1/12:0, D7-cholesterol, CE 17:0, MG 17:0, 4ME 16:0 diether DG, D5-TG 16:0/18:0/16:0 (Avanti Polar Lipids, Alabaster, AL). Relative molar amounts of lipid species in each sample were calculated based on appropriate class-based internal standards and presented as nmol. The data were then normalized to the sum of these molar contributions to obtain comparable relative contributions of each lipid species or class and expressed as mol% relative to total amount of lipidome measured. Relative molar amounts of lipid species in each sample were normalized per μg of total BAL protein and presented as nmol/μg of protein. BAL lipidomics data, including relative molar amounts of lipid species in each sample (nmol) and normalized data (nmol/μg protein) are provided in the Source Data file.

Lipid abbreviations are as follows: (Cer) Ceramide, (dhCer) Dihydroceramide, (SM) Sphingomyelin, (dhSM) Dihydrosphingomyelin, (MhCer) Monohexosylceramide (galactosylceramide + glucosylceramide), (Sulf) Sulfatide, (LacCer) Lactosylceramide, (GM3) Monosialodihexosylganglioside, (GB3) Globotriaosylceramide, (PA) Phosphatidic acid, (PC) Phosphatylcholine, (PCe) Ether phosphatidylcholine, (PE) Phosphatidylethanolamine, (PEp) Plasmalogen phosphatidylethanolamine, (PS) Phosphatidylserine, (PI) Phosphatidylinositol, (PG) Phosphatidylglycerol, (BMP) Bis(monoacylglycero) phosphate, (AcylPG) Acyl Phosphatidylglycerol, (LPC) Lysophosphatidylcholine, (LPCe) Ether lysophosphatidylcholine, (LPE) Lysophosphatidylethanolamine, (LPEp) Plasmogen Lysophosphatidylethanolamine, (LPI) Lysophosphatidylinositol, (LPS) Lysophosphatidylserine, (NAPE) N-Acyl Phosphatidylethanolamine, (NAPS) N-Acyl Phosphatidylserine, (NSer) N-Acyl Serine.

## Pulmonary cell isolation and culture

For isolation and quantification of pulmonary cells, fluorescent activated cell sorting (FACS) was employed using fresh lung preparations. Mouse lung digests for single-cell flow cytometry were prepared from *Lrat⁺/⁺* (wild-type C57BL/6 mice) and *Lrat⁻/⁻* mice using a method involving digestion with pronase and collagenase. Briefly, the lungs of anesthetized mice were first perfused through the right ventricle with calcium- and magnesium-free Hanks' balanced salt solution (HBSS). Then the lungs were perfused in situ with HBSS containing calcium, magnesium, and Dispase II (Sigma-Aldrich) followed by a second perfusion with HBSS containing calcium, magnesium, and type IV collagenase (Worthington). The lungs were removed, rinsed and minced into small pieces. The minced lung tissue was incubated in HBSS containing Dispase II (Sigma-Aldrich), type IV collagenase (Worthington), and DNase I (Sigma-Aldrich) on a rotating shaker maintained at 37 °C for 60 min. At each of three 20-min intervals, the minced tissue was then passed 10-times through a 10-mL pipette to dissociate cells. The resultant cell suspension was next passed through a 100 μm strainer to collect single cells. Cells were then isolated using either magnetic (MACS) or fluorescence activated cell sorting (FACS).

For MACS-based isolation of ECs, negative depletion with Cd45 microbeads (Miltenyi Biotec) and positive selection for the platelet endothelial cell adhesion molecule (PECAM-1/Cd31) by magnetic labeling with microbeads in MACS MS columns (Miltenyi Biotec) was employed as described in[116]. Alveolar epithelial cell purification was performed using negative depletion with Cd45 microbeads (Miltenyi Biotec) and positive selection for the epithelial-cell adhesion molecule (Ep-CAM/Cd326) by magnetic labeling with microbeads in MACS MS columns (Miltenyi Biotec) as described in[117].

For flow cytometry isolations, cells were incubated with FcBlock (Biolegend) and stained with a mixture of fluorochrome-conjugated antibodies (Supplementary Table 1) using dilution 1:200 for each antibody. Flow cytometry was employed to sort live, individual retinoid-containing cells based on their emission at 455 nm (UV-positive cells)[58]. Singlet discrimination was sequentially performed using plots for forward scatter (FSC-A versus FSC-H) and side scatter (SSC-W versus SSC-H). Dead cells were excluded by scatter characteristics and staining with propidium iodide. Among UV-positive cells, subsets of cells with high expression of Cd45 (immune cells), Cd326 (epithelial cells), and Cd31 (endothelial cells) were identified, separated, and collected for further analysis. All FACS isolations were performed on a FACSAria cell sorter (Becton Dickinson) at the Columbia Stem Cell

Initiative (CSCI) Flow Cytometry shared resource lab. FACS data were analyzed using FlowJo software version 10 (TreeStar, Ashland, OR).

After isolation from *Lrat*[+/+] (wild-type C57BL/6) lungs, a fraction of UV-positive cells was either collected for analysis (referred to as freshly isolated or day 0 cells) or plated on 6-well plastic plates at a density $2 \times 10^5$ cells/well followed by culture in DMEM (Life Technologies Corporation, Carlsbad, CA, USA) supplemented with 10% FBS (Peak Serum, Wellington, CO, USA), 100 units/mL penicillin and 100 μg/mL streptomycin (Life Technologies Corporation, Carlsbad, CA, USA) in a humidified 5% $CO_2$ incubator at 37 °C. Culture medium was first changed after overnight (12 h) culture and subsequently every 2 days.

For analysis of cultured cells, attached cells were washed with PBS, treated with 0.25% trypsin-EDTA solution (Life Technologies Corporation, Carlsbad, CA, USA) and then collected by centrifugation. Cells were harvested after 12 h culture (day 0.5 time point), and 3-day (72 h) culture (day 3 time point). The harvested cells were pelleted and snap frozen in liquid $N_2$ and stored at −80 °C until analysis. The DNA concentration of each cell sample was assessed employing a High Sensitivity Quant-iT™ dsDNA Assay Kit (Thermo Fischer Scientific, Waltham, MA, USA) according to the manufacturer's protocol.

### Single cell RNA-sequencing and bioinformatic analysis

UV-positive cells with viability >70% were isolated from *Lrat*[+/+] (wild-type C57BL/6) male mice (C57BL/6 genetic background) and submitted for single cell RNA sequencing (scRNA-seq) at The Single Cell Analysis Core at JP Sulzberger Columbia Genome Center. scRNA-seq experiments were done on two separate occasions and data generated from these analyses were then integrated (Supplementary Fig. 6B). Single cell libraries were prepared using the Chromium Single Cell 3' Reagent Kits v3.1 (10x Genomics) according to the manufacturer's instructions followed by their sequencing on a NovaSeq 6000 (Illumina) in a $2 \times 100$ bp configuration. Initial data processing, including demultiplexing reads and converting raw base call files into the fastq format, was performed using Cell Ranger version 5.0.1 (10X Genomics). Expression counts for each gene in all samples were collapsed and normalized to unique molecular identifier (UMI) counts using Cell Ranger 5.0.1 (10X Genomics). The result is a large digital expression matrix with cell barcodes as rows and gene identities as columns. Filtered feature matrix outputs from Cell Ranger were used for all downstream analyses.

Post-processing analysis was performed using the Seurat package version 4.0.4[91] in R Software version 4.1.1 (Vienna, Austria; https://www.r-project.org/). Expression profiles of cells from different samples were first analyzed and clustered separately. Before the analysis, quality control on each individual sample was undertaken on the number of genes detected in each cell (nFeature_RNA), number of transcripts detected in each cell (nCount_RNA), and percentage of mitochondrial (percent_mito) and ribosomal (percent_ribo) genes in each cell. These metrics were further used to exclude outliers, low quality cells with low gene numbers, or/and transcripts detected per cell and/or cells with a high number of transcripts mapped to mitochondrial genes. Cells were filtered using the *Subset* function to exclude all cells with <200 detected genes or >10% mitochondrial genes. The *NormalizeData* function was used to normalize and log scale the data. A subset of features that exhibit high cell-to-cell variation in the dataset was next calculated using the *FindVariableFeatures* function. The *ScaleData* function was used to regress out heterogeneity associated with cell cycle stage or mitochondrial contamination. To integrate the samples, integration anchors were identified using the *FindIntegrationAnchors* and the list of samples (individual seurat objects) were integrated using *IntegrateData* function. Canonical correlation analysis (CCA) and mutual nearest neighbors (MNN) detection functions in Seurat were used to remove batch-associated effects. Principle component analysis (PCA) on the scaled data was performed for dimension reduction using the *RunPCA* function. The

optimum dimensionality of the dataset for downstream clustering was determined using both the *JackStraw* and *ElbowPlot* methods. K-nearest neighbor (KNN) graph was constructed based on the Euclidean distance in PCA space, and the edge weights between any two cells based on the shared overlap in their local neighborhoods (Jaccard similarity) were refined using the *FindNeighbors* function. To cluster the cells, the Louvain algorithm was applied by using the *FindClusters* function at low resolution (resolution = 0.2). The cluster resolution value was determined using the *Clustree* function (Supplementary Fig. 6c) and biologically relevant cell clusters were defined as clusters that have at least 10 unique differentially expressed genes. Visualization of the clusters on a 2D map was performed with uniform manifold approximation and projection (UMAP) using *RunUMAP* (dims = 1:15). Differentially expressed genes of each cluster were identified using the Wilcoxon rank-sum test with the *FindAllMarkers* function.

A combination of supervised and unsupervised approaches to determine cell type annotation was employed. To identify major cell type subsets (stromal, epithelial, endothelial, and myeloid), we assessed the expression of the canonical markers *Col1a1*, *Epcam*, *Pecam 1*, and *Ptprc*. The *FindAllMarkers* function was used to identify a list of top genes that was unique to each cell cluster. To obtain the most sensitive and specific differentially expressed genes for each cluster, we identified genes with a *p*-value of <10^−5 and an average log fold-change >1. Preliminary annotation was undertaken by comparing this list to previously published conserved gene lists from single cell RNA sequencing data in mouse lung tissue[82–86]. Top conserved markers of final annotated cell types were visualized again with *FeaturePlot, ViolinPlot,* and *DotPlot*.

The stromal clusters were separated for further analysis using the *Subset* function. *FindNeighbors* and *FindClusters* (resolution = 0.2) functions were further applied to the fibroblast subset. The cluster resolution value was determined using the *Clustree* function (Supplementary Fig. 1f) and biologically relevant cell clusters were defined as clusters that have at least 10 unique differentially expressed genes. Visualization of the clusters on a 2D map was performed with uniform manifold approximation and projection (UMAP) using *RunUMAP* (dims = 1:15). Differentially expressed genes of each fibroblast cluster were identified using the Wilcoxon rank-sum test with the *FindAllMarkers* function. Top conserved markers of final annotated cell types were visualized again with *DotPlot*.

### RNA preparation and quantitative real time PCR (qRT-PCR)

Total RNA was extracted from frozen tissues or cell preparations employing TRIzol Reagent (Ambion, Foster City, CA, USA) and isolated using the E.Z.N.A Total RNA Kit II (Omega Bio-tek, Norcross, GA, USA) according to the manufacturer's protocol. RNA was quantitated at 260 nm using a Nanodrop spectrophotometer. cDNA synthesis was performed using 0.5 μg of total RNA (in a final volume of 20 μL) and was carried out for 10 min at 25 °C followed by 120 min at 37 °C employing reverse transcriptase (Applied Biosystems, Foster City, CA, USA). The reaction was stopped at 85 °C for 5 min, using a thermal cycler (Eppendorf, Westbury, NY, USA). The primers employed for qRT-PCR analyses of target gene expression are provided in Supplementary Table 2. 18 S RNA was employed as the reference housekeeping gene used to normalize mRNA expression.

qRT-PCR was performed in a total volume of 20 μl, including 40 ng of cDNA template, forward and reverse primers (100 nM each), and PerfeCTa SYBR Green FastMix (Quantabio, Beverly, MA, USA) using a LightCycler 480 instrument controlled by LightCycler 480 software version 1.5.0.39 (Roche, Branchburg, NJ, USA). After initial denaturation and enzyme activation (95 °C for 10 min), 40 cycles (94 °C for 10 sec, 55 °C for 30 sec, 72 °C for 30 sec) were performed for the annealing/extension steps, and fluorescence was measured. A dissociation curve program was performed after each cycle. Expression

levels of target genes were calculated based on the efficiency of each reaction and the crossing point deviation of each sample versus control and are expressed as fold differences in comparison with the reference gene.

## Western blotting

Lung and liver proteins were extracted using RIPA lysis and extraction buffer (Thermo Scientific) containing protease inhibitor cocktail (Millipore Sigma, Burlington, MA). For tissue and cell samples, 20 μg of total homogenate protein was analyzed. For plasma protein analysis, 2 μl of plasma was employed. The proteins were subjected to SDS PAGE in 12% gels followed by a transfer onto PVDF membrane. Immunoblotting was performed using primary rabbit polyclonal antibody against either mouse retinol-binding protein 4 (Rbp4, 1:3000), mouse lecithin:retinol acyltransferase (Lrat, 1:2000; Novus Biologicals), or a mouse monoclonal antibody against the *myc* epitope tag (Myc-tag, 1:2000; Invitrogen). Protein loading and relative quantification of protein expression were normalized using either β-actin (1:10,000; Millipore Sigma) or Gapdh (1:1000; Cell Signaling) as reference proteins. Secondary horseradish peroxidase-linked donkey anti-rabbit antibody (1:10,000; Thermo Fisher Scientific) or sheep anti-mouse antibody (1:10,000; GE Healthcare) were used accordingly. Protein bands were visualized using Super Signal West Pico PLUS chemiluminescence system (Thermo Scientific, Rockford, IL), membranes were digitally scanned using Odyssey XF Infrared Imager controlled by a LI-COR Acquisition Software version 1.0.19 (LI-COR, Inc., Lincoln, NE) and analyzed using Image Studio version 5.2 (LI-COR, Inc., Lincoln, NE) and ImageJ version 1.53a (National Institutes of Health) to generate quantitative relative protein expression data[118].

## Statistical analysis

All data are presented as means ± S.D. Student's *t*-test was used to analyze differences between genotype groups. Statistical differences involving more than two groups (*Lrat*[+/+], *Lrat*[−/−], LPS treatment with 5 mg/kg and 25 mg/kg, and 300 IU retinol treatment) were first analyzed by a one-way ANOVA followed by multiple comparisons employing Tukey's HSD *post hoc* test. Survival rates were analyzed by the Kaplan–Meier log rank test using GraphPad Prism version 9. *P*-values <0.05 were considered significant.

## Reporting summary

Further information on research design is available in the Nature Portfolio Reporting Summary linked to this article.

# Data availability

All scRNA sequencing data, including raw fastq sequencing files, gene expression matrices, and associated cell metadata generated in this study have been deposited in the NCBI's Gene Expression Omnibus (GEO) database under accession number GSE198521. Source data are provided with this paper.

# Code availability

Code used for the analysis of scRNA-seq data is available at the public GitHub repository at https://github.com/ishmarakov/scRNAseq_lung_retinoid.

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

## Acknowledgements

The authors wish to acknowledge financial support from grants R01DK068437, R01DK122071 and R21AA028110 (to W.S.B.) and grants R01HL36024, R01HL57556, and R01HL122730 (to J.B.) from the US Public Health Services, National Institutes of Health as well as support through the NIH/NCI Cancer Center Support Grant P30CA013696 allow the use of Columbia's Genomics and High Throughput Screening Shared Resource. M.M.-M. was funded by the EST19/00211 mobility grant from the Spanish Ministry of Universities. The authors would like to acknowledge the contributions of Dr. Christine Kim Garcia and Dr. Wellington Cardozo for many helpful discussions and for reading and commenting on the manuscript prior to its submission. We would also like to thank the staff of the Columbia Stem Cell Initiative Flow Cytometry core facility at Columbia University Irving Medical Center and the Biomarkers Core Laboratory at the Irving Institute of Clinical and Translational Research at Columbia University for their contributions to the work presented in this manuscript.

## Author contributions

I.O.S. wrote the first draft of the manuscript. I.O.S., G.A.G., M.N.I., and M.M.-M. carried out experiments, contributed to the design of experiments, discussed research plans and data, and helped write and edit the manuscript; and J.B. and W.S.B. contributed to the design of experiments, discussed research plans and data, helped write and edit the manuscript and provided financial support for the studies.

## Competing interests

The Authors declare no competing interests.
