## [Peer Review File · Nature Communications]

Retinoids Stored Locally in the Lung Are Required to Attenuate the Severity of Acute Lung Injury in Male MiceREVIEWER COMMENTS

Reviewer #1 (Remarks to the Author):

General:

Here, the authors demonstrate that:

- Lung RE concentration but not hepatic RE concentration decreases after LPS. Therefore, they conclude that the drop in lung RE concentration is not due to decreased production by the liver.
- Lung ROH and plasma ROH concentrations decrease after LPS. The plasma concentration of RBP4, the carrier for ROH, also declines after LPS, as does hepatic Rbp4 production.
- A knockout mouse that does not have pulmonary RE or ROH but does have 50% normal plasma ROH (LRAT KO) have increased mortality after LPS, whereas a mouse that has normal pulmonary RE and ROH but low circulating ROH has normal mortality rates. This suggests that the local pulmonary retinoid stores but not the circulating ROH levels are important for survival. There were significant alterations in surfactant-associated lipid species in the LRAT KO mice. There are also alterations in expression of tight and adherens junctions genes.
- Retinoid containing cells, purified by their autofluorescence in the UV channel, are mainly fibroblasts. As they differentiate into myofibroblasts in culture, they downregulate LRAT expression.
- Retinoid containing cells, purified by their autofluorescence in the UV channel, also include other cell types such as epithelial cells, endothelial cells, and immune cells, and these cells express many of the retinoid synthesis and uptake genes.
- There are extensive changes in pulmonary retinoids and in genes that regulate retinoid mobilization after LPS.
- Lethality in LRAT knockout mice was rescued with dietary retinoids.

Major Comments:

1. Based on the decline in plasma concentration of RBP4 and hepatic Rbp4 production, they conclude that the decline in plasma ROH was due to decreased RBP4 secretion from the liver rather than increased ROH utilization in the inflamed tissues. This conclusion is not justified. Causality has not been demonstrated.
 2. Moreover, the subsequent conclusion that the decline in lung ROH results from its direct use locally to buffer against the LPS insult is confusing. The authors should explain why a decline in lung ROH is not due to a decline in plasma ROH, in turn due to decreased liver RBP4. Furthermore, even if lung RE and ROH levels are declining independently of plasma and liver levels, it is not justified to conclude that that decline results from its "direct use locally to buffer against the LPS insult". Perhaps it is just being degraded in the lung rather than consumed for some protective purpose. Finally, the term "buffer" is nonspecific and should be explicitly defined.
 3. For the general reader, the authors should provide more background on the use of autofluorescence to purify retinoid-accumulating cells, including the specificity of this method and how it has been validated. The conclusion that the non-fibroblast cell types accumulate retinoids depends on the robustness of this method to purify exclusively retinoid-accumulating cells.
 4. Sftpa and Sftpb are not specific for AEC2s. Therefore, the conclusion that AEC2s accumulate retinoids is not justified.
 5. It is unclear why sc-RNAseq was necessary to identify the retinoid-accumulating cells. Instead, the cells could have been simply stained with various antibodies such as EpCAM, CD31, and CD45 to identify epithelial, endothelial, and immune populations by flow cytometry. If those were also UV+, the same conclusion could have been drawn. The expression of the retinoid uptake genes by these cell types could have easily been assessed from the many publically available lung sc-RNAseq datasets. Moreover, demonstrating the importance of retinoid accumulation by a non-lipofibroblast cell type after LPS would greatly strengthen the manuscript.
 6. The LRAT knockout mice have worse outcomes after LPS, which demonstrates that pulmonary retinoids are protective. There are significant alterations in surfactant lipid species and intercellular junction gene expressions. There are probably many other things that would be found to be altered if they were measured. However, the mechanism through which pulmonary retinoids protect the lung has not been demonstrated. Without this, the manuscript is largely descriptive.
- Minor Comments

1. A hypothesis should be stated.
2. The introduction is long and diffuse and includes information that is not relevant (such as emphysema trials of ATRA) and appears to omit information that is relevant (such as background about retinoids for the general reader -e.g., different species of retinoids, how they are related, how and where they are synthesized, how retinoids are made and secreted by the liver into the plasma, etc).
3. Figure 1 should be moved to the Supplement.
4. Fig. 2A,B would be easier to read if Lung PBS and LPS bars were next to each other, and liver PBS and LPS bars were next to each other.
5. In line 137, the authors explain that LRAT is responsible for RE synthesis, but then go on to state that ROH concentrations are decreased in LRAT knockout mice in line 138. The data show that both RE and ROH concentrations are decreased. The text should also state that RE concentrations are decreased. Moreover, the authors should explain why ROH concentrations are decrease in the LRAT knockout mice if LRAT is necessary for RE synthesis.
6. Plasma RE levels should be shown in Supplementary Fig. 3.

Reviewer #2 (Remarks to the Author):

The study by Shmarakov et al. reports numerous paradigm shifting findings related to the uptake, metabolism, and roles of retinoids during lung inflammation. The authors show very convincingly that LPS-triggered inflammation results in remodeling of retinoid metabolism in lungs (but not liver) towards increased uptake of retinol from plasma, increased hydrolysis of local pulmonary stores of retinyl esters to provide abundant retinol for retinoic acid (ATRA) biosynthesis and, as a result, increased production of ATRA. Through elegant use of animal models, the authors also show that the primary source of retinol for the lung are the intestinal lipoproteins that deliver retinol to lungs postprandially. This is in contrast to previous thinking that lungs obtain their retinoids from the circulating plasma RBP4-retinol secreted from liver. Thus, the local stores of retinoids in the lungs are critically important during inflammation, and the authors' data that these stores can be repleted through dietary vitamin A supplementation can have important clinical implications for treatment of inflammatory lung disease.

Moreover, the authors identify several distinct lung cell populations that have not been previously known to take up and store retinoids; in this regard, it is especially interesting that they are able to distinguish two populations of lipofibroblasts: the proliferative versus those that express markers of more differentiated state. The ATRA content could regulate the balance between these two states. The authors' data also suggest that different cell populations may potentially exchange their pools of retinoids.

There are several minor concerns that should be addressed.

The authors state that a more than 5-fold elevation in pulmonary ATRA was observed 24 hours after LPS (5 mg/kg) instillation (Fig. 6C). However, it appears that the increase is closer to 2-fold. It is stated that none of the cell clusters expressed the gene encoding stimulated by retinoic acid 6 (Stra6), but Fig 6 D shows an increase in STRA6. Which lung cells express Stra6?

Why are pulmonary RE and ROL lower in RBP4 KO after LPS?

RDH10 mRNA is increased ~5-fold upon LPS treatment. Have the authors checked the RDH10 protein levels and also DHRS3 mRNA and protein?

Reviewer #3 (Remarks to the Author):

This is a substantial and well-done study adding very interesting insights to the mechanisms how retinols are instrumental in limiting lung damage.

It would be great if the authors could describe in more detail how they did the retinol measurements in the UV-isolated cells (Figure 3C Supplementary Figure 4), e.g. information on number of cells necessary for meaningful measurements, total amounts of individual species measured, limits, of detection and of quantitation.

Also, retinyl linoleate and retinyl oleate peaks should be identified in Supplementary Figures 1 and 4.

Reviewer #4 (Remarks to the Author):

The study by Shmarakov et al. used single-cell RNA-seq to describe the role of local retinoid metabolism in different cell populations within the lungs in helping to limit acute lung injury. The study aims to explain the molecular interplay across different lung cells during acute lung injury and how it is influenced by local cellular retinoid storage. Generally, the study is interesting and robust technically. It offers interesting insights given the contributions of ATRA-RAR to several lung cell types. Below are some comments on the technical aspects of this submission:

1. With respect to Figure 2D, loading control appears to be missing in this figure. Please show loading control and normalized quantitation.
2. Apart from using *Lrat* and *Rbp4* knockout mice, have the authors considered using inhibitors of retinoid synthesis and signaling to deplete local stores of cellular retinoids to make the claim the local pulmonary retinoid stores are critical for mediating ALI?
3. Please clarify in line 193, if Figure 4A and supplementary figure 5 were done using PCA or UMAP, as figures appear to look like a UMAP instead of PCA. Both dimensional reduction methods are quite different.
4. With reference to Figure 4 and supplementary Figure 6, comparative analysis of transcriptional signatures from previous literature appear to show that cluster 0, 1 and 2 contain many subclusters of fibroblasts population. Therefore it is worth splitting individual clusters into further subclusters by increasing the cluster resolution parameter (See `seurat FindClusters`). Though proper annotation may be difficult across datasets from previous literature studies as discussed, Figure 4G still does not present an accurate representation of individual cells and since the non-fibroblast cells are removed, they should be re-clustered again differently.
5. Can the authors clarify further on justifying performing single-cell RNA-sequencing on UV-positive cells to bias the collection of retinoid-containing cells as opposed to general cell population from the lungs?

Minor comments:

1. Significant bars/markers could be added to Figures 2E and F for better clarity.
2. Missing scale bar and labels in Figure 3B. An arrow to indicate elongated morphology will help better clarify the text.
3. The scaling of the violin plots in Figure 4C and 4D look squished vertically, please provide a better resolution of the figure.

RESPONSES TO REVIEWER COMMENTS

We thank the Reviewers for their comments. The changes that we made to our manuscript in response to these comments, we believe, have considerably strengthened it.

Below, first, we provide a brief summary of all of the additions we have made to the revised manuscript.

Subsequently, we provide detailed responses to each of the four Reviewers' comments and references supporting our arguments. *We first restate verbatim the comments from each of the Reviewers, as we received them. This is done using a normal font. We then respond to each of the Reviewers' comments in bold italicized font. The specific changes made in the revised manuscript in response to each comment are described.*

Brief Summary of Additions Made to the Revised Manuscript in Response to Reviewer Comments

1. Generation and studies of a new mouse model.
 - By crossing a mouse line harboring a transgene encoding a dominant/negative retinoic acid receptor (Rardn) mutant protein (Rardn^{fl/fl} mice) with mice expressing Cre-recombinase (Cre) under the control of a tamoxifen-inducible EC-specific vascular endothelial cadherin (VE-Cad/Cdh5) promoter (EC-CreER^{T2} mice), we generated, and report findings obtained through study of a new mouse model expressing the Rardn transgene specifically in endothelial cells (EC-Rardn mice). The Rardn transgene disrupts all-*trans*-retinoic acid-retinoic acid receptor transcriptional regulation in vascular endothelial cells.
 - In **new Suppl. Figure 11** we provide immunoblot characterizations of Rardn transgene expression establishing cell-specific transgene expression in magnetic cell sorting-(MACS)-isolated primary lung endothelial cells (ECs).
 - In **new Figures 7A and 7B**, we provide *in vivo* survival data for LPS-challenged EC-Rardn mice.
2. Data obtained from new primary lung cell isolations and their molecular and functional characterization.
 - In **new Figures 2A and 2B** we provided data obtained from FACS analysis of isolated lung cells prepared from wild type (*Lrat*^{+/+}) and *Lrat*^{-/-} mice. This establishes the specificity of retinoid autofluorescence for use in isolating retinoid-containing cells.
 - In **new Suppl. Figure 7** we report new FACS analysis data involving the use of antigen-specific fluorochrome-labeled antibodies to identify myeloid (Cd45⁺), epithelial (Cd326⁺), and endothelial (Cd31⁺) cells among sorted UV-positive cells.
 - In **new Suppl. Figure 5** we provide analyses of retinoid levels present in the cell preparations collected by FACS sorting based on retinoid autofluorescence.

- In **new Figures 3E and 3F** we provide mRNA and protein expression analyses of the *Lrat* gene in whole lung homogenates, primary lung lipofibroblasts, and primary lung endothelial cells.
 - In **new Suppl. Figure 10** we report *Stra6* and *Tnfa* mRNA expression analyses in cells isolated from BAL fluids obtained from PBS- and LPS-treated wild-type (*Lrat*^{+/+}) mice.
3. We provide new post-processing analysis of scRNA-seq datasets containing either all lung UV-positive cells or a subset of UV-positive stromal cells.
- In **new Suppl. Figures 6C and 6E** we report a new clustering analyses employing different cluster resolutions for scRNA-seq datasets containing all lung UV-positive cells as well as a subset of UV-positive stromal cells.
 - In **new Figures 3D and 3G** we report post-processing analyses of our scRNA-seq datasets to assess the expression level of genes involved in retinoid metabolism and signaling across cell clusters of lung UV-positive cells.
 - In **new Suppl. Figure 6D** we report post-processing analyses of our scRNA-seq datasets using specific markers for alveolar epithelial type 2 cells (AEC2) and alveolar epithelial type 1 cells (AEC1).
4. Other new data from new analyses include:
- In **new Suppl. Figure 4B** we report new data from *Cyp26a1* and *Cyp26b1* mRNA expression analysis in the lungs of wild type (*Lrat*^{+/+}) and *Lrat*^{-/-} mice.
 - In **new Figure 6C** we report plasma retinol (ROH) concentrations for wild type (*Lrat*^{+/+}) and *Lrat*^{-/-} mice 24 hours after nasal instillation of PBS or LPS at a dose of 5 mg/kg.

Our Responses to Reviewer #1

General:

Here, the authors demonstrate that:

- Lung RE concentration but not hepatic RE concentration decreases after LPS. Therefore, they conclude that the drop in lung RE concentration is not due to decreased production by the liver.
- Lung ROH and plasma ROH concentrations decrease after LPS. The plasma concentration of RBP4, the carrier for ROH, also declines after LPS, as does hepatic Rbp4 production.
- A knockout mouse that does not have pulmonary RE or ROH but does have 50% normal plasma ROH (LRAT KO) have increased mortality after LPS, whereas a mouse that has normal pulmonary RE and ROH but low circulating ROH has normal mortality rates. This suggests that the local pulmonary retinoid stores but not the circulating ROH levels are important for survival. There were significant alterations in surfactant associated lipid species in the LRAT KO mice. There are also alterations in expression of tight and adherens junctions genes.

- Retinoid containing cells, purified by their autofluorescence in the UV channel, are mainly fibroblasts. As they differentiate into myofibroblasts in culture, they downregulate LRAT expression.
- Retinoid containing cells, purified by their autofluorescence in the UV channel, also include other cell types such as epithelial cells, endothelial cells, and immune cells, and these cells express many of the retinoid synthesis and uptake genes.
- There are extensive changes in pulmonary retinoids and in genes that regulate retinoid mobilization after LPS.
- Lethality in LRAT knockout mice was rescued with dietary retinoids.

Reviewer #1 summarizes well many of our findings. We agree fully with each item in the summary. We thank Review #1 for the careful read of the manuscript and the thoughtful comments below regarding our work. We believe that we have responded fully to these comments.

Major Comments:

1. Based on the decline in plasma concentration of RBP4 and hepatic Rbp4 production, they conclude that the decline in plasma ROH was due to decreased RBP4 secretion from the liver rather than increased ROH utilization in the inflamed tissues. This conclusion is not justified. Causality has not been demonstrated.

It is well established from the Global Health literature that infections and the associated inflammatory responses seen in humans result in a substantial lowering of plasma retinol and RBP4 levels (1-3). This observation also has been made in animal model studies (4-9). Rodent studies, including ones involving inflammation induced by LPS (5,6,8,9) have established that the observed decline arises primarily from diminished hepatic RBP4 synthesis and secretion (4-9). Since retinol can only be mobilized from the liver bound to RBP4, this also results in lower plasma retinol levels. This literature further establishes that neither renal filtration nor tissue uptake and catabolism contribute significantly to the observed decline in circulating retinol-RBP4. In the original manuscript, we simply based our consideration of this point on this literature but we failed to explicitly state this in the manuscript. We have now added this information to the revised text on new page 6, starting on new line 111.

However, as Reviewer #1 indicates, we have not specifically shown this for our experimental studies. Consequently, to address Reviewer #1's concern, we have modified the manuscript to be more cautious in how we consider this point. The revised text now suggests on new page 6, new line 121, that other possibilities also may account for this observation.

2. Moreover, the subsequent conclusion that the decline in lung ROH results from its direct use locally to buffer against the LPS insult is confusing. The authors should

explain why a decline in lung ROH is not due to a decline in plasma ROH, in turn due to decreased liver RBP4. Furthermore, even if lung RE and ROH levels are declining independently of plasma and liver levels, it is not justified to conclude that that decline results from its “direct use locally to buffer against the LPS insult”. Perhaps it is just being degraded in the lung rather than consumed for some protective purpose. Finally, the term “buffer” is nonspecific and should be explicitly defined.

We have partially explained the rationale for this reasoning, one that is based on an extensive literature, in our response above to Reviewer #1’s first comment. Additionally, we have revised the text of the manuscript in response to this second comment. We have removed completely old line 127-128 on old page 6 which stated “...use locally to buffer against...” from the revised manuscript.

Data in new Figure 6, Panel D (and originally provided in old Figure 6), shows a statistically significant increase in lung ATRA levels in wild type mice (mice which possess substantial local ROH and RE stores) upon administration by instillation of a dose of LPS given at 5 mg/kg. However, an increase in lung ATRA levels is not observed for matched *Lrat*^{-/-} mice, that lack ROH and RE stores, receiving the same LPS dose. Moreover, in (old and new) Figure 1 we show that neither ROH nor RE stores in the liver decline in response to an LPS challenge. In fact, liver ROH levels are significantly greater in wild type mice receiving the LPS challenge, an observation that is consistent with the notion that during an acute phase response the liver is secreting less ROH-RBP4. Finally, as seen in new Figure 6C, plasma ROH levels are not different for PBS- versus LPS-treated mice. Based on these observations, we propose that the lack of an increase in ATRA levels in *Lrat*^{-/-} animals in response to an LPS challenge arises due to the absence of precursor ROH and RE in the lungs of these animals. The inability to generate ATRA from local RE and ROH stores, we further propose, accounts for the poorer responses of *Lrat*^{-/-} mice to an LPS insult that induces acute lung injury.

As further proof that ATRA-RAR signaling is required to have a better response to LPS induce lung injury, we provide new data in new Figure 7 that supports this contention. Using a newly generated mouse model harboring a cell type-specific ablation of RAR-mediated ATRA signaling in vascular endothelial cells, we demonstrate that ATRA-RAR signaling in endothelial cells is required for maintaining survival in the face of high dose LPS-induced lung inflammation. These new data are now presented and discussed on new pages 25 and 26 (lines 544-568).

3. For the general reader, the authors should provide more background on the use of autofluorescence to purify retinoid-accumulating cells, including the specificity this method and how it has been validated. The conclusion that the non-fibroblast cell types accumulate retinoids depends on this robustness of this method to purify exclusively retinoid-accumulating cells.

The autofluorescence of cellular retinoid stores was first described in the early 1980s (10-12). This autofluorescence is very distinct and unusual, since it involves excitation at 350 nm but the characteristic blue-green emission occurs at 450-460 nm. This unusual characteristic (specifically, the considerable distance between the excitation and emission maxima) for this autofluorescence provides confidence in the use of FACS to isolate retinoid-storing cells. Retinoid autofluorescence has been used in published studies to establish the identities of retinoid-storing cells and/or to quantitate the abundance of these cells (13-16). We (17) and others (18-21) have used this characteristic to isolate by FACS and study retinoid-storing cells from living tissues. This approach is now increasingly being used in published studies to isolate retinoid-containing cells (17-21).

To address more fully the reviewer's comment regarding the specificity and robustness of this methodology, we have undertaken additional new validating experiments that we now present in new Figure 2 and discuss on page 9 (lines 192-194). Data from this experiment convincingly shows that for *Lrat*^{-/-} animals, with no RE present in their lungs, no UV-positive cells can be detected when we use the same FACS gating strategy that we used to detect and isolate UV-positive cells from wild type (*Lrat*^{+/+}) mice. This confirms that the retinoid stores present within lungs are responsible for the autofluorescence of these cells.

We further note that for the studies reported in the revised manuscript, the FACS isolated non-fibroblast cell types that possess endogenous retinoid autofluorescence also strongly express LRAT and other gene products like *Rbp1* that are needed for facilitating retinoid storage. We point out that the sole known physiological role of LRAT is to synthesize retinyl esters and the sole post-absorptive function of retinyl esters is for retinoid storage. Thus, these non-fibroblast cell types not only possess endogenous retinoid autofluorescence, they also possess the metabolic machinery needed to allow for retinoid accumulation. This is a very novel finding from our study that substantially changes understanding of how retinoid uptake and storage take place in tissues.

As requested by Reviewer #1, we have added new text on new page 8, on new lines 175-183, to provide for readers more background on these issues.

We had originally hypothesized that vascular endothelial cells play an important role in the uptake of retinoid from the circulation prior to transferring it to parenchymal cells. We wanted to understand the molecular basis for this. But, never-the-less, we were very surprised by the finding that non-fibroblast cell types accumulate retinoids. This was unanticipated by the literature. Consequently, this finding is an important one that very significantly extends understanding of cellular retinoid metabolism and actions locally in tissues where retinoic acid-dependent transcriptional regulation is required.

4. Sftpa and Sftpb are not specific for AEC2s. Therefore, the conclusion that AEC2s accumulate retinoids is not justified.

In response to this comment regarding the specificity and the proper annotation of UV-positive epithelial cells found in our scRNA-seq data sets, we have applied additional cell-specific markers described in the literature to our analysis. These new data are now presented in the new Suppl. Figure 6 and discussed on new page 11, on lines 228-237. We conclude in the text based on our analysis these new data “..., these cells did not express other markers of AEC1 late differentiation⁶⁵⁻⁶⁷ like Pdpn, Cav1, Hopx, Scnn1g, and Scnn1b (Suppl. Fig. 6D). This allowed us to annotate the cells in this cluster as predominantly AEC2s, both canonical AEC2s and transitional AEC2s that are in the process of differentiation towards AEC1s⁶⁶.”

5. It is unclear why scRNA-seq was necessary to identify the retinoid-accumulating cells. Instead, the cells could have been simply stained with various antibodies such as EpCAM, CD31, and CD45 to identify epithelial, endothelial, and immune populations by flow cytometry. If those were also UV+, the same conclusion could have been drawn. The expression of the retinoid uptake genes by these cell types could have easily been assessed from the many publically available lung sc-RNAseq datasets. Moreover, demonstrating the importance of retinoid accumulation by a nonlipofibroblast cell type after LPS would greatly strengthen the manuscript.

We understand and appreciate Reviewer #1’s point. However, when we originally planned our scRNA-seq experiment, we expected to identify a variety of different retinoid-containing fibroblasts (UV-positive cells containing retinoids---as discussed above) in our data sets. None of the existing scRNA-seq databases provide information specifically regarding cellular retinoid content. We did not expect to identify any cell types other than fibroblasts as being retinoid-containing. Therefore, we could not predict at the onset of our studies the need for various cell-specific antibodies to isolate other cell types via flow cytometry. To our surprise, not only fibroblasts were identified in our data sets of UV-positive cells, but also endothelial, epithelial, and myeloid cells. This unexpected discovery underscores the novelty and paradigm-shifting nature of the data generated by our study.

Nonetheless, as a note of added proof, we have undertaken additional new experiments as suggested by Reviewer #1. We isolated primary lung cells using cell-specific antibodies, including EpCAM, CD31, and CD45 to identify epithelial, endothelial, and immune cells among the UV-positive cells by FACS. These new data are now presented in new Suppl. Figure 7 and discussed on new page 11. Lines 240-248. We believe that inclusion of these new data in the revised manuscript further strengthens our work, manuscript, and conclusions.

We also agree with the reviewer regarding the demonstration of the importance of retinoid accumulation by a nonfibroblast cell type. To address this comment and further strengthen our manuscript, we have undertaken a new study involving a newly generated mouse line with ablated RAR-mediated ATRA signaling in endothelial cells. These new data are now presented in new Figure 7 and discussed on new page 25, lines 544-568.

We were originally interested in understanding the processes through which vascular endothelial cells act in facilitating retinoid uptake from the blood for use to regulate transcription in the parenchymal cells composing tissues. We chose to use the lung as the tissue model for our studies since lung vascular endothelial cells are relatively well studied and the lung both accumulates and requires retinoids for the maintenance of its health. To some degree, we were also motivated to study lung since we were initiating this work coincidentally with the arrival of COVID-19.

We also note that our review of existing public scRNA-seq databases, as highlighted in new Suppl. Figure 8, indicated that there is not complete concurrence between these databases. We noted this lack of concurrence between databases in the original manuscript on old page 23, starting on old line 511, citing old references 61-65. However, we did not stress this point in the original manuscript. Thus, we chose to undertake new scRNA-seq determinations for our studies since this provided us with unconflicted (“clean”) data, generated in our lab, that could be used to formulate both new ideas and new questions regarding the retinoid-related questions that are of interest to us. We prefer to continue to leave this issue understated in our revised text, although new Suppl. Figure 8 clearly speak to this issue.

6. The LRAT knockout mice have worse outcomes after LPS, which demonstrates that pulmonary retinoids are protective. There are significant alterations in surfactant lipid species and intercellular junction gene expressions. There are probably many other things that would be found to be altered if they were measured. However, the mechanism through which pulmonary retinoids protect the lung has not been demonstrated. Without this, the manuscript is largely descriptive.

We agree with Reviewer #1 that retinoids are needed to maintain many functions within the cells of the alveolar niche. As evidenced by Figure 3, Panel G, the 3 RARs are expressed at some level in all of these cells. In keeping with the reviewer’s comment, to establish the cellular need for ATRA-RAR signaling in endothelial cells, we now report findings from a new study of mice where we have specifically ablated RAR signaling in endothelial cells. These new data (as new Figure 7, Panel B, discussed on page 25, lines 544-568) provide mechanistic understanding of a critical role of intact ATRA-RAR signaling in endothelial cells for survival during ALI. As is well understood, the lung endothelial barrier constitutes the critical protective mechanism against lung inflammation and

injury. These new data show that endothelial overexpression of a dominant-negative mutant RAR form that blocks ATRA signaling specifically in endothelial cells compromises the barrier, increasing mortality. This novel finding mechanistically implicates endothelial ATRA in lung endothelial barrier protection during severe lung inflammation. To our knowledge, this revealed role of ATRA is outstandingly novel and requires further understanding as to how ATRA impacts multiple components of the barrier through detailed studies that are outside the present scope.

Minor Comments

1. A hypothesis should be stated.

Our original hypothesis, based on our preliminary and published work as well as published work by others, was that vascular endothelial cells within retinoid-responsive tissues like the lung play a very significant and unrecognized role in the uptake and metabolism of retinoids. Our new data have confirmed this hypothesis. Moreover, our new data clearly show the critical role of uncompromised ATRA-RAR signaling in endothelial cells for survival during acute lung injury. This had not been previously recognized by the literature. As requested by Reviewer # 1, we now state this hypothesis in the second paragraph of the revised Introduction on new page 4, new lines 64-67.

2. The introduction is long and diffuse and includes information that is not relevant (such as emphysema trials of ATRA) and appears to omit information that is relevant (such as background about retinoids for the general reader -e.g., different species of retinoids, how they are related, how and where they are synthesized, how retinoids are made and secreted by the liver into the plasma, etc).

In response to this comment, we have revised the Introduction to shorten it (reduced the text from 47 total lines to 32) and to make the text less diffuse. To accomplish this, we have removed much of the original text that constituted the old first, third and fourth paragraphs of the old Introduction. As suggested by the Reviewer, we also have added new text to the old second paragraph to better explain natural retinoid chemical forms, their relationships to each other and their metabolism. We also have added a metabolic scheme as Panel A of new Suppl. Figure 1. This will benefit reader understanding of retinoid metabolism in the new text.

3. Figure 1 should be moved to the Supplement.

This has been done. Old Figure 1, is now provided in Suppl. Figure 1, Panel B.

4. Fig. 2A,B would be easier to read if Lung PBS and LPS bars were next to each other, and liver PBS and LPS bars were next to each other.

These changes have been made in the revised version of the manuscript.

5. In line 137, the authors explain that LRAT is responsible for RE synthesis, but then go on to state that ROH concentrations are decreased in LRAT knockout mice in line 138. The data show that both RE and ROH concentrations are decreased. The text should also state that RE concentrations are decreased. Moreover, the authors should explain why ROH concentrations are decrease in the LRAT knockout mice if LRAT is necessary for RE synthesis.

We have clarified our discussion of retinoid metabolism in $Lrat^{-/-}$ mice and modified the text to avoid this confusion. We have also more strongly supported our explanation with the addition new data presented in new Suppl. Fig. 4. The short answer to this question is the ROH is being drawn away for ATRA synthesis and by subsequent ATRA catabolism. LRAT is proposed in the literature to provide a mechanism for trapping retinol as retinyl ester that can later be used for ATRA synthesis.

6. Plasma RE levels should be shown in Supplementary Fig. 3.

We do not present data on plasma RE levels. This is because of the absence of any detectable REs in the fasting plasma obtained from mice used in our experiments.

Our Responses to Reviewer #2

The study by Shmarakov et al. reports numerous paradigm shifting findings related to the uptake, metabolism, and roles of retinoids during lung inflammation. The authors show very convincingly that LPS-triggered inflammation results in remodeling of retinoid metabolism in lungs (but not liver) towards increased uptake of retinol from plasma, increased hydrolysis of local pulmonary stores of retinyl esters to provide abundant retinol for retinoic acid (ATRA) biosynthesis and, as a result, increased production of ATRA. Through elegant use of animal models, the authors also show that the primary source of retinol for the lung are the intestinal lipoproteins that deliver retinol to lungs postprandially. This is in contrast to previous thinking that lungs obtain their retinoids from the circulating plasma RBP4-retinol secreted from liver. Thus, the local stores of retinoids in the lungs are critically important during inflammation, and the authors' data that these stores can be replenished through dietary vitamin A supplementation can have important clinical implications for treatment of inflammatory lung disease.

Moreover, the authors identify several distinct lung cell populations that have not been previously known to take up and store retinoids; in this regard, it is especially interesting that they are able to distinguish two populations of lipofibroblasts: the proliferative versus those that express markers of more differentiated state. The ATRA content could regulate the balance between these two states. The authors' data also suggest that different cell populations may potentially exchange their pools of retinoids.

We thank Reviewer #2 for the careful read and thoughtful review of our manuscript.

There are several minor concerns that should be addressed.

The authors state that a more than 5-fold elevation in pulmonary ATRA was observed 24 hours after LPS (5 mg/kg) instillation (Fig. 6C). However, it appears that the increase is closer to 2-fold.

Reviewer # 2 is correct. We have corrected this in the revised manuscript.

It is stated that none of the cell clusters expressed the gene encoding stimulated by retinoic acid 6 (Stra6), but Fig 6 D shows an increase in STRA6. Which lung cells express Stra6?

The reviewer is correct in pointing out that while none of the cells from the scRNA-seq data set express Stra6, we never-the-less report an upregulation in Stra6 expression in the whole lung tissue during acute lung injury. New data presented in new Suppl. Figure 10 suggest that the elevated levels of Stra6 transcription is brought about by inflammatory cells that infiltrate the lungs upon LPS-induced lung injury. This is now discussed on new page 23 (lines 507-511).

Why are pulmonary RE and ROL lower in RBP4 KO after LPS?

RDH10 mRNA is increased ~5-fold upon LPS treatment. Have the authors checked the RDH10 protein levels and also DHRS3 mRNA and protein?

We thank the reviewer for pointing out to this interesting observation to which we have not paid sufficient attention. We have been focused on understanding the observed lethality phenotype seen in the *Lrat*^{-/-} mice and consequently paid less attention to the *Rbp4*^{-/-} animals. We do not have proper data to address directly this question and we can only speculate on why this is the case. The answer to this question may be directly associated with the previous comment regarding the lung *Rbp4*-*Stra6* axis during lung inflammation. As we have stated in the manuscript, we believe that upregulated *Stra6* expression in *Rbp4*^{-/-} animals does not result in an increase of ROH uptake from ROH:RBP4 complex and therefore locally stored retinoids are used after LPS-induced lung injury.

We have not checked protein expression for Rdh10, nor have we checked mRNA and protein expression DHRS3. However, we have reanalyzed our scRNA-seq data sets and identified DHRS3 as one of the most highly expressed genes involved in retinoid metabolism in pulmonary retinoid-containing UV-positive cells. These new data are now presented in new Figure 3, Panel 3D, starting on new page 12, lines 266-274.

Our Responses to Reviewer #3

This is a substantial and well-done study adding very interesting insights to the mechanisms how retinols are instrumental in limiting lung damage.

We thank Reviewer #3 for the careful reading of our manuscript and for comments.

It would be great if the authors could describe in more detail how they did the retinol measurements in the UV-isolated cells (Figure 3C Supplementary Figure 4), e.g. information on number of cells necessary for meaningful measurements, total amounts of individual species measured, limits, of detection and of quantitation.

In response to Reviewer #3's comment, we have added this information to the revised manuscript.

We totally agree with the reviewer that the amounts of intracellular retinoids being analyzed by HPLC in the UV-isolated cell preparations are very low and this could raise a question regarding the accuracy of the performed quantitative measurements. We want to emphasize that the majority of HPLC analyses of the cells, including the peaks and spectral characteristics, were acquired to show the presence of retinoids (mainly REs) in the isolated cell preparations. We are aware that some of these quantities may not allow for accurate quantification of intracellular retinoid concentrations using standard HPLC methodologies.

Because of the very low intracellular concentrations, we were not able to measure quantitatively ROH in our FACS-isolated lung UV-positive cell preparations. Although we have indicated a retention time for the ROH peak on the chromatogram (Suppl. Fig. 5A). Moreover, we were able to integrate RE peaks and provide quantitative measurements for intracellular RE concentrations in isolated UV-positive cells. For these measurements, we utilized about 300-400 thousand cells from one lung and normalized RE concentrations per ng of DNA (Fig. 2D). However, in our new experiment, when we isolated labeled Cd326⁺ (epithelial), Cd31⁺ (endothelial), and Cd45⁺ (immune) cells from UV-positive cells by flow cytometry, we were able to collect about 15-25 thousand cells for each cell type from the pool of UV-positive cells. These cell preparations were further used to assess for the presence of RE and these new data are presented in new Suppl. Fig. 5. Although we were able to detect the peaks corresponding to REs in

each isolated cell preparation, we were unable to analyze them quantitatively because of their low concentrations.

Also, retinyl linoleate and retinyl oleate peaks should be identified in Supplementary Figures 1 and 4.

In response to this comment, we have annotated the peaks corresponding to retinyl linoleate and retinyl oleate in Suppl. Figures 2 and 5.

Our Responses to Reviewer #4

The study by Shmarakov et al. used single-cell RNA-seq to describe the role of local retinoid metabolism in different cell populations within the lungs in helping to limit acute lung injury. The study aims to explain the molecular interplay across different lung cells during acute lung injury and how it is influenced by local cellular retinoid storage. Generally, the study is interesting and robust technically. It offers interesting insights given the contributions of ATRA-RAR to several lung cell types. Below are some comments on the technical aspects of this submission:

We thank Reviewer #4 for the careful review of our manuscript and for the thoughtful comments.

1. With respect to Figure 2D, loading control appears to be missing in this figure. Please show loading control and normalized quantitation.

We understand the reviewer's point. We have normalized the protein load by adding an equal amount of plasma (2 ul) to each lane in order to detect relative plasma RBP4 content by immunoblot. This is a standard approach used in the literature and in our previous publications. Some examples of this include:

1 Yang, Q. et al. Serum retinol binding protein 4 contributes to insulin resistance in obesity and type 2 diabetes. Nature 436, 356-362, doi:10.1038/nature03711 (2005).

2 Lee, S. A. et al. Adipocyte-specific overexpression of retinol-binding protein 4 causes hepatic steatosis in mice. Hepatology 64, 1534-1546, doi:10.1002/hep.28659 (2016).

Moreover, we maintain that RBP4 content normalized by plasma volume is likely more accurate than normalizing to other plasma protein(s), whose concentration(s), like those of RBP4, can be affected during acute lung inflammation as well.

2. Apart from using Lrat and Rbp4 knockout mice, have the authors considered using inhibitors of retinoid synthesis and signaling to deplete local stores of cellular retinoids to make the claim the local pulmonary retinoid stores are critical for mediating ALI?

We totally agree with the reviewer's point and are planning to expand our future studies in understanding the role of retinoids in lung disease by using approaches involving inhibition of retinoid synthesis and signaling. As a step forward in this direction and to partially address the reviewer's comment, we have undertaken new studies involving a newly created mouse line with ablated RAR-mediated ATRA signaling in endothelial cells (EC-Rardn mice). These new data are now presented in new Figure 7 and discussed on new page 25 (lines 544-568)

3. Please clarify in line 193, if Figure 4A and supplementary figure 5 were done using PCA or UMAP, as figures appear to look like a UMAP instead of PCA. Both dimensional reduction methods are quite different.

We thank the reviewer for picking up this discrepancy and our poor editing. We have modified the text to be correct on this point.

4. With reference to Figure 4 and supplementary Figure 6, comparative analysis of transcriptional signatures from previous literature appear to show that cluster 0, 1 and 2 contain many subclusters of fibroblasts population. Therefore, it is worth splitting individual clusters into further subclusters by increasing the cluster resolution parameter (See Seurat FindClusters). Though proper annotation may be difficult across datasets from previous literature studies as discussed, Figure 4G still does not present an accurate representation of individual cells and since the non-fibroblast cells are removed, they should be re-clustered again differently.

We appreciate Reviewer #4's point. Consequently, we have undertaken a new comparative analysis of our RNA-seq data sets with ones provided in the literature to properly annotate cells in our data set. However, as discussed in the revised manuscript and also pointed out by the reviewer, we were not able to use in full the existing marker gene sets available in the published literature to annotate fibroblast cell clusters in our data set. Therefore, new Figure 5 and new Suppl. Figure 8 are only intended to emphasize this discrepancy and are not aimed at showing many subclusters of the fibroblast population. In fact, these figures show that the same cell cluster can have several different names depending on the publication and the authors' preferences. In our initial postprocessing analysis, we used different cluster resolution parameters ranging from 0.2 to 0.9. We are now providing these cluster trees as new Suppl. Figure 6C and 6E. At the same time, we were cautious about "excessive" clustering that

may have technical, but not biological relevance. That's why we have determined biologically relevant cell clusters as the ones having at least 10 unique differentially expressed genes. Because we were not able to meet this criterion with a cluster resolution of more than 0.2, we undertook further manipulations using this resolution.

We agree with the Reviewer's point regarding the need to re-cluster the fibroblast clusters after removing all non-fibroblast clusters. These data are now presented in new Figure 4A. Here, again we used different cluster resolution parameters ranging from 0.2 to 0.9 (this is now presented in new Suppl. Figure 6E). And again, we choose to proceed with the same 0.2 resolution in our further analysis to be on a safe side of annotating biologically relevant cell clusters. This is because we were not able to have all of the clusters with at least 10 unique differentially expressed genes when we applied cluster resolution 0.3. Re-clustering of the stromal clusters resulted in a different appearance of the cell groups in UMAP. However, this yielded the same differentially expressed gene patterns.

We plan to further apply scRNA-seq methodology to understand the biology of pulmonary fibroblasts and the role that retinoid accumulation has in defining their transcriptional signatures and phenotype. As a part of these future studies, we will isolate and sequence pulmonary fibroblasts based on the differences in their retinoid autofluorescence (and therefore retinoid content) as well as using fluorescent protein labels specifically expressed by pulmonary fibroblasts.

5. Can the authors clarify further on justifying performing single-cell RNA-sequencing on UV-positive cells to bias the collection of retinoid-containing cells as opposed to general cell population from the lungs?

Reviewer #1 also raised a question (Comment 5) along these lines. Please see our responses to Reviewer #1 for more details. As we indicated in our responses to Reviewer #1, we originally were interested in understanding the processes by which vascular endothelial cells act in facilitating retinoid uptake from the blood for transfer to and use by the parenchymal cells composing tissues in order to regulate transcription. We also were interested in understanding retinoid storage in the lungs. Information regarding retinoid levels within the lungs is unavailable in existing scRNA-seq databases. Consequently, we chose to focus on UV-positive cells collected by FACS and analyzed by scRNA-seq. We chose to use the lung as a tissue model for our studies since lung vascular endothelial cells are relatively well studied and the lung both accumulates and requires vitamin A for the maintenance of its health. To some degree, we were also motivated to study lung since we were initiating this work coincidentally with the arrival of COVID-19. Our initial review of existing public sc-RNAseq databases indicated that there is not complete concurrence between these public databases. We noted this lack of concurrence between databases in the original manuscript on

old page 23 (new page 28), starting on old line 511 (starting on new line 623), citing references 61-65 (new references 82-86), however we did not stress this point. Thus, we chose to undertake new sc-RNAseq analyses for our studies since this would provide us with unconflicted (“clean”) data, generated in our lab, that allowed us to formulate both new ideas and new question regarding the retinoid-related questions of interest to us.

We specifically chose to isolate retinoid-containing cells by FACS since it is this population of cells that accumulates the circulating retinoid taken up and transferred by the vascular endothelial cells. Initially, we had expected that we would identify processes responsible for transferring circulating retinoid from the blood to retinoid-storing cells. Moreover, we originally expected only one cell type, the lipofibroblasts, would be found to accumulate retinoid and we wanted to gain understanding of gene expression patterns in these cells that might allow for this. Instead, surprisingly, our data established that both vascular endothelial cells and pulmonary epithelial cells accumulate retinoid stores and express genes encoding proteins that are required for retinoid accumulation, transport and metabolism.

In response to this comment and the one by Reviewer #1, we now indicate this on new page 11, lines 240-248.

Minor comments:

1. Significant bars/markers could be added to Figures 2E and F for better clarity.

We have added P-values to Figures 1E and 1F for the Kaplan-Meier curves when statistical significance was detected.

2. Missing scale bar and labels in Figure 3B. An arrow to indicate elongated morphology will help better clarify the text.

The micrographs of FACS isolated lung retinoid-containing cells were taken at 40X magnification. We have added scale bars, dashed arrows to Figure 2C to indicate elongated morphology and solid arrows to indicate lipid droplets.

3. The scaling of the violin plots in Figure 4C and 4D look squished vertically, please provide a better resolution of the figure.

We have redone the violin plots as suggested by the reviewer to provide better resolution in the figures.

REFERENCES SUPPORTING OUR RESPONSES

1. Stephensen CB. 2001. Vitamin A, infection, and immune function. **Annu. Rev. Nutr.** 21:167-192,
2. Rubin LP, Ross AC, Stephensen CB, Bohn T, and Tanumihardjo SA. 2017. Metabolic Effects of Inflammation on Vitamin A and Carotenoids in Humans and Animal Models. **Adv. Nutr.** 21:197-212.
3. Rosales FJ, and Ross AC. 1996. Inflammation in Human Immunodeficiency Virus Type I Infection as a Cause of Decreased Plasma Retinol. **J. Infect. Dis.** 173:507-508.
4. Felding P, and Fex G. 1985. Rates of synthesis of prealbumin and retinol-binding protein during acute inflammation in the rat. **Acta Physiol. Scand.** 123:477-483.
5. Rosales FJ, Ritter SJ, Zolfaghari R, Smith JE, and Ross AC. 1996. Effects of acute inflammation on plasma retinol, retinol-binding protein, and its mRNA in the liver and kidneys of vitamin A-sufficient rats. **J. Lipid Res.** 37:962-971.
6. Rosales FJ, and Ross AC. 1998. Acute inflammation induces hyporetinemia and modifies the plasma and tissue response to vitamin A supplementation in marginally vitamin A-deficient rats. **J. Nutr.** 128:960-966.
7. Gieng SH, Raila J, and Rosales FJ. 2005. Accumulation of retinol in the liver after prolonged hyporetinolemia in the vitamin A-sufficient rat. **J. Lipid Res.** 46:641-649.
8. Cieng SH, Green MH, Green JB, and Rosales FJ. 2007. Model-based compartmental analysis indicates a reduced mobilization of hepatic vitamin A during inflammation in rats. **J. Lipid. Res.** 48:904-912.
9. Wu L, and Ross AC. 2013. Inflammation induced by lipopolysaccharide does not prevent the vitamin A and retinoic-induced increased in retinyl ester formation in neonatal rat lungs. **J. Nutr.** 109:1739-1745.
10. Wake K. 1980. Perisinusoidal stellate cells (fat-storing cells, interstitial cells, lipocytes), their related structure in and around the liver sinusoids, and vitamin A-storing cells in extrahepatic organs. **Int. Rev. Cytol.** 66:303-353.
11. Senoo H, Kojima N, and Sato M. 2007. Vitamin A-Storing Cells (Stellate Cells). **Vitamins and Hormones** 75:131-159.
12. Senoo H, Yoshikawa K, Morii M, Miura M, Imai K, and Mezaki Y. 2010. Hepatic stellate cell (vitamin A-storing cell) and its relative – past, present and future. **Cell Biol. Int.** 34:1247-1272.
13. Nagy NE, Holven KB, Roos N, Senoo H, Kojima N, Norum KR, and Blomhoff R. 1997. Storage of vitamin A in extrahepatic stellate cells in normal rats. **J. Lipid Res.** 38:645-658.
14. Thompson KC, Trowern A, Fowell A, Marathe M, Haycock C, Arthur MJP, and Sheron N. 1998. Primary Rat and Mouse Hepatic Stellate Cells Express the Macrophage Inhibitor Cytokine Interleukin-10 During the Course of Activation In Vitro. **Hepatology** 28:1518-1524.
15. Zhang XY, Sun CK and Wheatley AM. 2000. A Novel Approach to the Quantification of Hepatic Stellate Cells in Intravital Fluorescence Microscopy of the Liver Using a Computerized Image Analysis System. **Microvascular Res.** 60:232-240.

16. Kida Y, Asahina K, Inoue K, Kawada N, Yoshizato K, Wake K, and Sato T. 2007. Characterization of vitamin A-storing cells in mouse fibrous kidneys using *Cygb*/STAP as a marker of activated stellate cells. **Arch. Histol. Cytol.** 70:95-106.
17. D'Ambrosio DN, Walewski JL, Clugston RD, Berk PD, Rippe RA, and Blaner WS. 2011. Distinct populations of hepatic stellate cells in the mouse liver have different capacities for retinoid and lipid storage. **PLoS One.** 6(9):e24993.
18. Ogawa T, Tateno C, Asahina K, Fujii H, Kawada N, Obara M, and Yoshizato K. 2007. Identification of vitamin A-free cells in a stellate cell-enriched fraction of normal rat liver as myofibroblasts. **Histochem. Cell. Biol.** 127:161-174.
19. Tacke F, Weiskirchen R. 2012. Update on hepatic stellate cells: pathogenic role in liver fibrosis and novel isolation techniques. **Expert Rev. Gastroenterol. Hepatol.** 6(1):67-80.
20. Bartneck M, Warzecha KT, Tag CG, Sauer-Lehnen S, Heymann F, Trautwein C, Weiskirchen R, and Tacke F. 2015. Isolation and Time Lapse Microscopy of Highly Pure Hepatic Stellate Cells. **Anal. Cell. Path.** Article ID 417023.
21. Balaphas A, Meyer J, Gameiro C, Frobert Am Giraud M-N, Egger B, Bühler LH, and Gonelle-Gispert C. 2022. Optimized Isolation and Characterization of C57BL/6 Mouse Hepatic Stellate Cells. **Cells.** 11:1379.

REVIEWERS' COMMENTS

Reviewer #1 (Remarks to the Author):

Here, the authors demonstrate that multiple lung cell types store retinoids and that lung retinoids are protective against lung injury and that endothelial cell retinoids are protective during homeostasis and injury. The authors have made many important additions to the manuscript. Supplemental Fig. 1 is helpful. The endothelial cell specific ATRA receptor loss of function model significantly strengthens the manuscript. The new data validating the use of autofluorescence to isolate retinoid + cells and associated explanation is very helpful for readers outside the retinoid field. The new flow data using cell type specific flow antibodies to identify retinol+ epithelial, endothelial, and myeloid cells strengthens the manuscript. There is only one minor statement that must be edited in lines 235-237:

Mature murine AEC2s express Aqp5 and Ager, so detecting these transcripts does not suggest that there are transitional AEC2s in the process of differentiation towards AEC1s. In fact, there are very, very few if any transitional AEC2s in the resting murine lung. The phrase “both canonical AEC2s and transitional AEC2s that are in the process of differentiation towards AEC1s” should be removed from the manuscript.

Reviewer #2 (Remarks to the Author):

The authors fully addressed my comments.

Reviewer #3 (Remarks to the Author):

Thank you for considering my comments for the revised version of the paper - this answers my (few) questions.

Reviewer #4 (Remarks to the Author):

The authors have addressed my comments adequately.

RESPONSES TO REVIEWER COMMENTS

We thank the Reviewers for their comments. The changes that we made to our manuscript in response to these comments, we believe, have considerably strengthened it.

Below, we provide a response to the comment raised by Reviewer #1.

Reviewer #1 (Remarks to the Author):

Here, the authors demonstrate that multiple lung cell types store retinoids and that lung retinoids are protective against lung injury and that endothelial cell retinoids are protective during homeostasis and injury. The authors have made many important additions to the manuscript. Supplemental Fig. 1 is helpful. The endothelial cell specific ATRA receptor loss of function model significantly strengthens the manuscript. The new data validating the use of autofluorescence to isolate retinoid + cells and associated explanation is very helpful for readers outside the retinoid field. The new flow data using cell type specific flow antibodies to identify retinol+ epithelial, endothelial, and myeloid cells strengthens the manuscript. There is only one minor statement that must be edited in lines 235-237: Mature murine AEC2s express Aqp5 and Ager, so detecting these transcripts does not suggest that there are transitional AEC2s in the process of differentiation towards AEC1s. In fact, there are very, very few if any transitional AEC2s in the resting murine lung. The phrase “both canonical AEC2s and transitional AEC2s that are in the process of differentiation towards AEC1s” should be removed from the manuscript.

Our Response to Reviewer #1

Reviewer #1 summarizes well the many important additions that we added to the revised manuscript. We thank Review #1 for the careful reading of our manuscript and the thoughtful comments regarding our work. We fully agree with the reviewer’s suggestion and have removed our incorrect statement in original lines 235-237.